# Persistent effects of sand extraction on habitats and associated benthic communities in the German Bight

Finn Mielck[1], Rune Michaelis[1], H. Christian Hass[1], Sarah Hertel[1], Caroline Ganal[2], Werner Armonies[1]

[1]Alfred Wegener Institute, Helmholtz Centre for Polar and Marine Research, Wadden Sea Research Station List auf Sylt, 25992, Germany
[2]Institute of Hydraulic Engineering and Water Resources Management, RWTH Aachen University, 52056, Germany

*Correspondence to:* Finn Mielck (finn.mielck@awi.de)

**Abstract.** Sea-level rise demands for protection measures of endangered coastlines crucial for the local population. At the island of Sylt in the SE North Sea, shoreline erosion is compensated by replenishment with sand dredged from an offshore extraction site. We studied the long-term effects of sand extraction on bathymetry, geomorphology, habitats, and benthic fauna. Sand extraction created dredging holes about 1 km in diameter and up to 20 m below the ambient seafloor level. Directly after dredging the superficial sediment layer inside the pits was dominated by coarse sand and stones. Hydroacoustic surveys revealed only minor changes of bathymetry >35 years after sand extraction. Obviously, backfill of the dredging pits was very slow, at a rate of a few mm per year, presumably resulting from low ambient sediment availability and relatively calm hydrodynamic conditions despite high wave energy during storms. Thus, a complete backfill of the deep extraction sites is likely to take centuries in this area. Hydroacoustic surveys and ground truthing showed that the backfilled material is mainly very fine sand and mud, turning the previously coarse sand surface into a muddy habitat. Accordingly, grab samples revealed significant differences in macrozoobenthos community composition, abundance and species density between recently dredged areas (<10 years ago), recovery sites (dredging activity >10 years ago) and undisturbed sites (control sites). Overall, dredging turned the original association of sand-dwelling species into a muddy sediment association. Since re-establishment of disturbed benthic communities depends on previous re-establishment of habitat characteristics, the low sedimentation rates indicate that a return to a pre-dredging habitat type with its former benthic community and habitat characteristics is likely to be impossible. Since coarse sand is virtually immobile in this area, a regeneration towards natural conditions is unlikely without human interference (e.g. mitigation measures like depositing coarse material on the seafloor to restore the sessile epifauna) or without a new ice age which once formed this area.

## 1 Introduction

Sea-level rise, with ever-increasing rates in the near future, demands protection measures of endangered coastlines crucial for the local population (Nicholls and Tol, 2006; Hinkel et al., 2014). In many cases, ecological awareness and sustainability considerations have led to the preference for 'soft' coastal protection measures like beach nourishment over 'hard' protection approaches such as dikes or revetments (Hamm et al., 2002; Pranzini et al., 2015; Staudt et al., 2020). As a result, there is a

world-wide high demand for marine aggregates needed especially for coastal protection. It has reached a high level on a worldwide scale and further increase is expected (Hamm et al., 2002; Kubicki, 2007a; Danovaro et al., 2018; Schoonees et al., 2019). For the northern European continental shelf, the extracted volume rose from altogether 53 million $m^3$ between 1998 and 2002 to a total of 73.2 million $m^3$ in 2018 (ICES, 2016, ICES, 2019).

Marine sand extraction changes local bathymetry and sediment composition (De Jong, 2016b, Mielck et al., 2018) and affects macrozoobenthic communities, both directly by killing or removal of benthic organisms during sediment extraction, and indirectly by altering the environmental conditions (Boyd et al., 2003; Foden et al., 2009). Further indirect effects of sediment dredging include increased turbidity, release of nutrients and toxins, changes in regional morphodynamics (Le Bot et al., 2010) and smothering of organisms due to sedimentation (van Rijn et al., 2004). Current attempts to minimize the area affected by dredging activities have led to greater extraction depths. However, the ecological effects of deep sand extraction (>10 m dredging depth) is still largely unknown (Boyd and Rees, 2003; De Jong et al., 2016a). Since sedimentological investigations showed tremendous change of the physical habitats, it must be expected that macrozoobenthic communities change at a similar level (Boyd et al., 2005; Kubicki et al., 2007b; Foden et al., 2010). Whether or not the benthic communities are able to recover to their pre-dredging state, remain disturbed, or new habitats with altered benthic communities are developed, is crucial information for a holistic assessment of the impact of such a coastal defense measure. It is thus essential to investigate the benthic communities of the affected areas to predict changes in species abundances and the structure of the benthic community.

After sediment extraction, morphological recovery of the local environment depends on the ambient sediment availability and hydrodynamic conditions. Additional crucial factors are extraction depth (i.e. deep drilling vs. shallow dredging) and the amount of material extracted (Cooper et al., 2007; De Jong et al., 2015). Re-establishment of the benthic community depends on the progress in morphological recovery and on the sensitivity and resilience of the different benthic organisms and communities to anthropogenic impact (Desprez, 2000; Cooper et al., 2011). In general, a full re-establishment of benthic assemblages is possible but may take a long time and is strongly depending on sediment composition, original topography and the connection to similar habitats in the proximity (Desprez, 2000; Boyd et al. 2005). In addition, recovery may proceed over intermediate stages atypical for the original environment, e.g. when large amounts of fine materials are deposited in a sandy area (Boers, 2005).

The aim of this study was to further follow the re-filling process of the dredging pits and, as a new aspect, to find out whether and how extensive marine aggregate extraction affects regional macrozoobenthic communities. If local faunal composition was mainly ruled by larval supply, faunal composition inside the dredging pits may be similar to ambient sediments. Otherwise, if sediment composition was an important factor, faunal composition in the muddy sediments of dredging pits should considerably differ from composition in ambient sandy sediments.

Thus, the main objectives were to (i) gain a deeper understanding of the correlation between the prevailing habitats and the associated benthic assemblages, to (ii) evaluate temporal recovery patterns along with short- and long-term changes in the community structures and to (iii) investigate the potential of a re-establishment to pre-dredging conditions. Therefore, dredging

pits of different ages and, as a control, the sandy areas surrounding the extraction site were compared for sediment and benthic faunal composition. Using hydroacoustics and sediment grab sampling, maps were created showing the sediment characteristics and morphology of the seafloor.

Hydroacoustics have proven very effective for remote sensing seafloor classification and habitat mapping. Multibeam echosounders give information about water depth and morphology, and can thus be used to calculate backfill rates at the extraction pits (Harris and Baker, 2012; Jones et al. 2016; Mielck et al., 2018). Sonar systems such as sidescan sonars allow to investigate the backscatter intensity by transmitting an acoustic pulse, which will be reflected by the seafloor and received by a transceiver. Backscatter allows to distinguish between hard/coarse (strong backscatter response from the seafloor) and

soft/fine substrates (low backscatter response from the seafloor (Blondel and Murton, 1997; Blondel, 2003; Mielck et al., 2012, Mielck et al., 2015)), which is an additional parameter useful for seafloor classification. Interpretation and verification of sonar data always require ground truthing, e.g. by sediment samples (for granulometry) and/or underwater video (Harris and Baker, 2012; Hass et al., 2017).

Benthos communities often largely correlate with sediment composition; however, a precise identification of communities is
not yet possible by hydroacoustic methods because transitional zones between major habitat types may be populated by transitional communities and these zones are often not detectable by hydroacoustic methods (Markert et al., 2013). Thus, ground truthing by sediment samples is required to correctly identify the benthic communities.

## 2 Materials and methods

### 2.1 Study Area

The study area "Westerland Dredging Area" (WDA) is located in the German Bight (SE North Sea) approx. 7 km west off the island of Sylt (Fig.1). This island suffers strong erosion, notably along its wave-exposed western side. Since 1972, sediment losses are compensated by artificial beach nourishments and the investigated study site serves as a sand extraction area since 1984 (LKN-SH, 2012). Most of the seafloor west off Sylt is covered with Holocene fine sand (Figge, 1981; Zeiler et al., 2000).
However, for shore nourishments coarse-to-medium grained Pleistocene sands outcropping in WDA are preferred (Temmler, 1983; 1994). These Pleistocene sediments come with gravel and stones deposited as a moraine core during the Saalian glaciation (~300–126 kyr BP). At the study area, this moraine core strikes in west-north-west direction (Köster, 1979, see Fig. 1). The surface of the seafloor in WDA is characterized by bands of coarse-grained rippled sand, so called sorted bedforms which are very common west off Sylt (Diesing et al., 2006; Mielck et al., 2018). Most of these bands have a wavelength of
~100 m and strike in east-west direction. The ripples within the coarse-grained areas do not strike in the same direction as they were most likely formed during storm events (alignment perpendicular to storm direction). The sorted bedforms are often

overlaid by a layer of migrating fine sand that leads to the consequence that their shape is frequently altered (Mielck et al. 2015).

The study area has an extent of ~5 km in north-south direction and ~3 km in east-west direction. Natural water depths range between ~14 and ~17.5 m while the pits left by sand extraction may reach down to 30 m water depth with diameters of approx. 1 km. Since 1984, more than 40 million m³ sediment was extracted from this area using trailing suction hopper dredgers (LKN-SH, 2012, 2020). With an actual annual material withdraw of 1–2 million m³, this area is the largest offshore sediment extraction site in Germany. The study area includes recent dredging zones (younger than 10 years), sand deposits exploited already more than 10 years ago, and unaffected seafloor regions. Meanwhile, the pits persist for more than 30 years (Mielck et al., 2018). The Pleistocene coarse sands in the pits exposed during sand extraction were rapidly covered by a layer of fine sand due to slides at the steep slopes. The fine sand originates from the immediate seafloor surface around the pits (Zeiler et al., 2004; Mielck et al., 2018). After this initial phase, muddy sediments accumulated, however, due to the combination of a lack of mobile sediments and low transport rates (Valerius et al., 2015), at very low rates only. Accordingly, a complete backfill of the deep dredging pits was estimated to take many decades (Mielck et al., 2018).

This research paper is a follow-up to the previous study by Mielck et al. 2018, which focused on morphological changes due to marine aggregate extraction in WDA using bathymetric data collected between 1993 and 2017. For the study presented here, hydroacoustic data and sediment samples were taken using the research vessel *Alkor* in January 2019. In order to acquire over-all information on the prevailing morphology and high-resolution backscatter data of the study area, altogether 55 transects, each 5.5 km long, with a lateral distance of 50 m were surveyed in north-south direction at a vessel speed of ~5 knots. During the survey, which took place between January 25[th] and January 27[th] at calm weather conditions, multibeam echosounder and sidescan sonars were used simultaneously on all transects. Subsequently, 53 grab samples for grain-size and macrobenthic faunal analyses were collected on January 31[th]. Surveyed transects and position of the grab samples are provided in Fig. 2. Underwater videos could not be acquired as a consequence of high turbidity.

## 2.2 Multibeam echosounder

Bathymetric information of the investigation area was collected using a shallow water multibeam echosounder SeaBeam 1180 (180 kHz; swath width of 150°) which was installed on a plate in the ships' moonpool. Positioning and motion compensation was done using a Kongsberg SEATEX MRU-Z. During the survey three CTD-profiles were measured (conductivity, temperature, pressure) to calculate sound velocities. Multibeam data were post-processed using Hypack 2016a and ESRI ArcGIS10 resulting in a bathymetric map with a grid size of 2 m. For tidal correction, the gauge "Westerland Messpfahl" was used, which is located approx. 6 km east of WDA. Depth values in this study are given in meters below mean sea level.

## 2.3 Sidescan sonar

Two different sidescan sonars were deployed simultaneously to determine backscatter properties (roughness) of the seafloor across the study area during the survey. The devices were attached to each other and towed behind the vessel to avoid sound disturbances from the ship. They operated with different frequencies in order to collect backscatter information from the seafloor in two resolutions, which provides more detailed data regarding sediment composition and habitat characteristics. The first sidescan sonar (Imagenex YellowFin 872) worked with a frequency of 330 kHz resulting in a resolution of 12.5 cm/pixel in the digital imaging while reaching a swath of 160 m on the seafloor. The second sidescan sonar was a Tritech StarFish 990F that operated with a frequency of 1 MHz and reached a resolution of ~1 cm/pixel at a swath of 60 m. Using different frequencies leads to more detailed information on the seafloor environment. Sidescan sonar data recorded with a low frequency generally yield information on large-scale objects on the seafloor (e.g. facies changes, sandwaves, megaripples) while a high frequency gives more information on small-scale structures such as ripple marks or stones (Mielck et al. 2015). All recorded sidescan sonar data were post-processed using SonarWiz 5 (Chesapeake Technology) resulting in a grid resolution of 0.5 m for the YellowFin and 5 cm for the StarFish. Distinct areas (e.g. fine/coarse sand) and characteristic backscatter responses in the sonograms (e.g. stones) were manually digitized using ArcGIS. The size of the stones was determined by measuring slant angle and length of the acoustic shadow using the software EdgeTech Discover. The results of the grain-size analysis were used to relate the backscatter intensities to the prevailing sediment distribution.

## 2.4 Grab sampling and analysis

The surface sediments and morphology across WDA are already well-known from the prior study (Mielck et al. 2018) and were taken as representative for all occurring seafloor environments. For ground truthing of hydroacoustic data and for macrobenthos analysis, a total of 53 grab samples were taken using a van Veen grab (HELCOM; 30 x 30 cm; 0.1 m²). The sampling positions generally followed a regular grid but some positions were also selected on the basis of the bathymetric information in order to take samples both from the older dredging pits (older than 10 years) and the newer ones (see Fig. 1). At two positions, sampling was not possible due to very steep slopes or the presence of stones on the seafloor, respectively, that prevented the sampler to close completely. Grain-size analyses were done using a CILAS 1180L diffraction laser particle-size analyzer, which provides grain-size information between 0.04 and 2500 µm. The statistical parameters (referring to vol-%) are based on Folk and Ward (1957) and were calculated using GRADISTAT (Blott and Pye 2001).

For faunal analyses, a sub-sample of 100 cm² surface area (max. depth of 18 cm; limited by the Van Veen grab) from each of the grabs was fixed in 5% buffered formaldehyde-in-seawater solution. For faunal analyses this sample volume may be unusually small but we judged it sufficient to find out whether dredging had a strong effect. In the lab, the sample was sieved through 1 mm square meshes and the residual fauna determined to species level and counted. Biomass was determined as wet weight per species and sample. For statistical analysis, the sampling sites were classified according to their history of sand

extraction: Class "0" for sites never impacted by sand extraction and thus serving as a control for undisturbed conditions; class "1" for the sites where sediment was extracted during the past 10 years; and class "2" for the sites where sand extraction terminated at least 10 years prior to sampling (cf. Fig. 2A). These classes were used as a categorical variable in univariate analyses of variances (ANOVAs) to test for effects on macrozoobenthic abundance, biomass, and species density. Significant differences between the variables were further investigated with Scheffe's post hoc test. Prior to statistical analyses, abundance and biomass data were $\log(x+1)$-transformed while Cochran C test indicated that no transformation was needed for species numbers. All calculations were done using STATISTICA® 6.1 software. All benthos data and results of the statistical analyses are included in the supplementary materials.

# 3 Results

## 3.1 Habitat mapping

The hydroacoustic survey executed in January 2019 revealed that all of the past dredging pits are still visible by bathymetric lows down to 30 m water depth (Fig. 2 left, multibeam echosounder measurements) and the pits of the various periods are still distinguishable from each other. The pits in the middle part of the study area are produced by dredging since 2017, the western ones from the 2009 to 2016 period, the southern ones from 1995 to 2008, and the northern ones from 1984 onwards (Fig. 2). Thus, even the oldest depressions have only partially refilled with sediment after 35 years (quick backfill of about 5 m and very low sedimentation rates after the first year; c.f. Zeiler et al. 2004; Mielck et al. 2018).

The sidescan sonar measurements (Fig. 2, right) showed numerous features across the study area (Fig. 3). Based on these measurements, the seafloor could be classified into four types (Fig. 3):

(1) Ground truthing with grain-size analyses of the sediment (Fig. 4A and B) revealed that high backscatter domains represent rippled coarse sand zones (sorted bedforms). Intermediate backscatter stands for fine sand. Coarse and fine sand zones were often demarcated by sharp borders (Fig. 3A). These backscatter patterns distinctly coinciding with the topography (compare Fig. 2 left and right), while low dunes composed of relatively mobile fine sand (extending in east-west direction) alternating with shallow troughs where relative immobile coarser sediment is exposed. These seafloor features are most pronounced on the undisturbed seafloor around the dredging pits, but also appear to extend into the older and shallower pits especially in the North of the study area.

(2) Several thousands of stones with diameters from ~10 cm to >1 m (best seen in the high-resolution data set, (Fig. 3B) occurred within this rippled coarse sand zone while there were virtually no stones present in the fine sand zones or dredging pits (Fig. 4C and D). Stones in sidescan sonograms are characterized by a strong dark reflection followed by a bright acoustic shadow.

(3) Extended areas of mud, in the sonograms represented as areas with uniform low backscatter, could only be identified in the dredging pits in the northern and in southern parts of WDA (Fig. 3C).

(4) In the center of the study area, where sand extraction is still ongoing, cone-shaped depressions were observed in the sonograms, which were caused by recent dredging activities (Fig. 3D).

The sediment distribution and bathymetric maps (Fig. 4) represent the spatial arrangement of these features in the studied area. While undisturbed ambient sediments were mostly fine sands interspersed with strips of coarse sand, the bottom of the holes left by sand extraction was characterized by coarse sands that were rapidly covered by a layer of fine sand during the first

month and later by muddy sediments. Sediments accumulating in >10-year-old pits were mainly fine and very fine sand with a mud content still significantly higher than in ambient sediments (Fig. 5).

## 3.2 Benthos analysis from grab samples

Sand extraction significantly changed macrozoobenthic abundance and species density while there was no significant effect

on biomass (ANOVA, Table 1). Scheffe post-hoc tests revealed that macrozoobenthic abundance was significantly lower in the dredged compared to the undisturbed sites ($p < 0.01$ for the recently dredged sites and $p < 0.05$ for the recovery sites) while there was no significant difference between recently dredged and recovery sites ($p = 0.53$; Fig. 5). After >10 years of recovery, the number of species returned to a level as high as for the control site ($p = 0.10$), while it was significantly lower in the recently dredged sites when compared to the undisturbed (control) site ($p < 0.01$; Fig. 5). These changes in macrozoobenthic species

density and abundance were accompanied by significant changes in sediment composition. Recently dredged sites had a low sand and a high mud content and were very heterogeneous sorted; older pit sediments were intermediate between fresh pits and ambient sediment, with intermediate sorting but with a mud content still far above ambient sediment level and with a median grain size only slightly higher than in fresh pits (Fig. 5). The percentage mud content differed significantly between all combinations of disturbance classes ($p < 0.05$) and therefore, may be best suited as a proxy for changes in sediment

composition.

Paralleling these changes in sediment composition, the composition of the macrozoobenthic community strongly changed during the recovery phase. Six of the ten most abundant species showed significant abundance variations between site classes (Table 2). Compared to the ambient sediments, abundance of the polychaetes *Magelona johnstoni, Pisione remota, Aonides paucibranchiata, Polygordius appendiculatus,* and *Goniadella bobretzkii*, all sand dwelling species, sharply dropped in fresh

(class 1) dredging holes and did not return to ambient levels in older pits (Table 2). The polychaete Nephtys cirrosa temporarily vanished from fresh dredging holes while mud-dwelling Notomastus latericeus temporarily showed up in the fresh pits. Older (class 2) dredging holes showed increases of abundance above ambient level in brittle stars *Ophiura ophiura* and its associate bivalve *Kurtiella bidentata*. Finally, the trumpet worm *Lagis koreni* became the numerical dominant species in older pits (Table 2). Most of these changes are likely to be caused by the changes in sediment composition: species that correlate

significantly with increasing median grain size decreased in the pits while the muddy-sand species *Lagis koreni* increased (Table 2). Thus, faunal composition in older dredging holes is an assemblage of muddy-sand dwellers and strongly differs

from the ambient assemblage of sand dwellers. A community composition equivalent to ambient conditions was not reached in any of the extraction pits.

## 4 Discussion

The potential for natural recovery of the seafloor morphology after sediment dredging depends on local sediment availability, the hydrodynamic conditions that determine sediment transport and sedimentation rates, and the extraction procedure (Desperez, 2000; Cooper et al., 2011; Goncalves et al., 2014; De Jong et al., 2015). A recovery of the benthic fauna in addition depends on the character of the newly accumulated material as well as on the sensibility and recruitment behavior of the involved benthic species (De Jong, 2016b).

For the sand mining area west off Sylt, hydroacoustic surveys and sediment analyses revealed that the impact of dredging on the seafloor morphology persists since many decades. Before the dredging activity started in 1984, the study site was characterized by patterns of fine and coarse sand (sorted bedforms, coinciding with seabed relief features), which are very common in this area (e.g. Figge, 1981, Mielck et al., 2015). These pre-dredging conditions are still present between the dredged areas and east of them (Fig. 2, 3A, 4) and seem to extend into the older and shallower dredging pits especially in the Northeast.

The dredging pits, in contrast, have different surface layers. Directly after dredging, the surface is composed of coarse sand and stones, that were too large to be sucked in by the dredger. Soon afterwards, this layer got more and more covered by fine sand probably deriving from the (formerly steep) rims of the pits (Zeiler et al., 2004; Mielck et al., 2018). Finally, the strong decrease of current velocities inside the pits allowed for sedimentation of suspended mud (Zeiler et al., 2004) turning the pits into mud areas after a couple of months. However, sedimentation rates are typically low in the southern North Sea and the

study area (Dominik et al., 1978; von Haugwitz et al., 1988; Mielck et al., 2018), which is brought about by the combination of a lack of mobile sediments and weak transport rates (Valerius et al., 2015). Therefore, mud accretion is a very slow process. The comparison of the 2019's bathymetry of the oldest pits with earlier measurements in 2016 and 2017 (Mielck et al., 2018) revealed no significant change indicating that the annual sedimentation rate was below the resolution of our multibeam device (~10 cm). This is in accordance with the very low sedimentation rate (2–18 mm per year) recorded from a muddy depression

near the Island of Helgoland, ~80 km south of the study area (Dominik et al., 1978; von Haugwitz et al., 1988). Based on such low rates of sedimentation, a complete backfill of the pits is likely to take centuries (Mielck et al., 2018). After refill, the previous accumulations of muddy material in deeper layers of the sediment will persist, potentially affecting the living conditions for deeper-dwelling fauna.

    This natural backfill cannot restore full pre-dredging conditions, for two other reasons: The first are differences in sediment

composition. While coarse-to-medium sand was removed during dredging, the backfill material is fine sand with a high mud content. This is due to the relative immobility of coarse sand (Tabat, 1979; Werner, 2004; Mielck et al., 2015).

The second reason relates to the numerous stones found in the undisturbed coarse sand areas. These are – as well as the coarse sand itself - natural relicts of Pleistocene moraines (Köster, 1979; Zeiler et al., 2008) highly unlikely to be transported by tidal currents. However, they provide the only natural hard substrates in a soft-sediment environment, giving a habitat to some sessile species and serving as stepping-stones in the dispersal of others (Sheehan et al., 2015; Michaelis et al., 2019). During sand mining, stones >10 cm are filtered out and remain on the seafloor (LKN-SH, pers. com., 2019). However, virtually no stones could be detected in the older dredging pits (Fig. 4 (c), (d)), as they were already buried by slope failures shortly after the dredging activity (Mielck et al. 2018). Thus, these patches of hard substrata and also the coarse sand areas are inevitably lost for the benthic epifauna. These habitats could only be restored by mitigation measures like depositing stones, gravel and coarse sand on the seafloor to allow for colonization of sessile epifauna.

When planning the study, benthic fauna was included as an additional aspect in the pit recovery process because previous studies (Mielck et al. 2018) indicated that the sediments accumulating in the pits were finer than ambient sediments, and small subsamples for macrobenthos were deemed sufficient to check whether change in sediment composition had a strong effect on benthic macrofauna that would justify further studies. Significant differences of faunal composition between dredged sites and ambient sediments despite the small sample volumes indicates that changes are indeed strong, both on the community and on the species level. Larger samples might have proven this for even more species.

Species typical for coarse grained sand such as the polychaetes *Pisione remota*, and *Polygordius appendiculatus* could not re-establish in the fine sediments of the pits because they have an interstitial life-style equivalent to meiofaunal-sized organisms, i.e. they need a sediment type with pore sizes large enough for movement without displacing the sand grains. Based on the realized distribution across sediment types in the eastern North Sea, most benthic species seem to be restricted to a species-specific spectrum of sediment composition (Armonies, in prep.). However, since sediment composition correlates with many other factors such as hydrodynamic stress and sediment stability (Snelgrove and Butman, 1994), or oxygen supply and biogeochemistry (Giere 2008, Giere et al. 1988), the causes for these restrictions are not clear. Therefore, we can only state that the sediment types occupied in the dredging pits coincide with the sediment types occupied in the surroundings (see supplementary material Table 2). In this sense, species composition of the benthic infauna changed according to sediment composition in the dredging holes.

Generally, recovery of the benthic fauna at disturbed sites depends on the recovery state of the sediment, and complete recovery is only possible if the native sediment characteristics are restored (Zeiler et al., 2004). Thus, complete recovery is only possible within the restrictions given above for the habitat characteristics. Until then, the original sandy habitat is lost for the benthic infauna, and thus as a feeding ground for higher trophic levels such as fish or diving birds that depend on sandy grounds or are limited by water depth. It is replaced by a new habitat type with a shallow muddy surface layer on top of a sandy sub-surface layer. This allows some surface-dwelling mud-fauna like *Lagis koreni* to come in (unless hampered by oxygen depletion brought about by restricted water circulation in the pits), but still excludes deep-dwelling mud-fauna such as *Callianassa subterranea* occurring in the muddy depression near the Island of Helgoland mentioned above. It may take some further

decades of mud accumulation to reach habitat characteristics comparable to the Helgoland depression, but only if the dredging pits will continue to act as sediment traps for muddy material.

Currently, sand mining accompanied by local habitat loss is restricted to a relatively small part of the SE North Sea with vast surrounding areas with similar habitats and fauna. Because of deep dredging operations instead of extensive dredging, a vast habitat loss is therefore not expected and not a threat to all the sand-dwelling benthic species living in the area as it was the

case in other marine areas (Varriale et al. 1985; Borja et al. 2006).

Instead, the deep mining pits provide local spots of muddy sediment, which is among the rarer habitat types in the SE North Sea. Judgement of prevailing pros and cons therefore depends on the item in focus but should always include that sand mining is just one of many types of anthropogenic exploitation in the area. Since faunal composition largely correlates with sediment composition, hydroacoustic habitat mapping is suggested as a cost-effective monitoring approach for the further development

of the extraction sites. Though, at present, hydroacoustic mapping cannot detect the full range of benthic habitats (e.g. in transition zones, Markert et al., 2013) it can indicate structural differences large enough to activate additional faunal studies.

As a strategy to monitor the further development in the extraction sites, we suggest investigations of the occurring habitat types by hydroacoustic means combined with the analysis of the benthic communities. For younger pits with fast rates of change, this should be done twice a year for habitat types and every two years for benthic fauna; for older pits with a slow rate of

change, a habitat survey every two years and a faunal analysis once per decade may be sufficient, to save money, time and resources.

Besides the fauna, mud accretion in the dredging pits may also affect the chemical environment. Mud often shows enriched contents of polycyclic aromatic hydrocarbons (PAHs), chlorine hydrocarbons (Brockmeyer and Theobald, 2016) or heavy metals (Lakhan et al., 2003). In addition, hydrodynamic conditions allowing for mud accretion might also facilitate

microplastic deposition. Whether or not the deep dredging pits seems to act as a sink for pollutants (Zeiler et al. 2004) and whether or not the pollutants affect the benthic fauna remains to be studied.

**5 Conclusion**

In the study area west off the island of Sylt (SE North Sea) the seafloor is characterized by a mix of fine and coarse sand patterns with occasional occurrences of stones. Sand extraction started in 1984 and created extraction pits about 1 km in

diameter and up to 20 m depth below ambient seafloor level. These mining pits remained virtually unchanged even after 35 years, with low rates of backfill by muddy sediment. The change in sediment composition from sand to mud caused changes in benthic community composition, turning the previous community of sand dwellers into a mud-preferring assemblage. Further development into a typical mud community may take some more decades, until the mud layer has become thick enough for deep-dwelling species. This state may then remain for the next centuries, until the pits are largely backfilled and attain a

surface sediment layer similar to original - at least regarding the morphology. But even then, living conditions may deviate

from the former conditions, because the fine backfill sediments changed the habitat permanently. In addition, stones, gravel and coarse sand originally occurring at the sediment surface are unlikely to be replaced; without human interference their function as a habitat for epibenthic species is inevitably lost. However, at some positions, especially in the flat pits in the Northeast, slight regeneration towards pre-dredging conditions becomes visible. Here, patterns of sediment, which are

coinciding with the seabed relief, recaptured the seafloor. This should be monitored in the future.

**Author contribution.** FM, HCH, WA designed the scientific study. FM, SH and CG collected the data during the research survey AL-519. SH, WA and FM processed and analyzed the data. FM, WA, RM and HCH prepared the manuscript.

**Competing interests.** The authors declare that they have no conflict of interest.

**Acknowledgements.** This study was funded by the German Federal Ministry of Education and Research (BMBF) and is part

of the joint research project STENCIL (Strategies and Tools for Environment-friendly Shore Nourishment as Climate Change Impact Low-Regret Measures; contract no. 03F0761), a collaborative coastal and shelf research program which aims to make further steps towards the establishment of an sustainable Integrated Coastal Zone Management (ICZM) and an Ecosystem Approach to Management (EAM) in Germany. We would like to acknowledge the master and crew of the research ship *Alkor* for supporting us during the surveys. Thanks are also due to the GEOMAR Kiel/Germany for providing us their multibeam

echosounder device during the survey AL-519.

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

*Table 1: Univariate ANOVAs for macrozoobenthos parameters versus site classes (undisturbed control / disturbance >10 years ago / disturbance during past 10 years). SSQ sum of squares, DF degrees of freedom, MSQ mean square, F F-statistic.*


| Parameter | SSQ | DF | MSQ | F | p |
|---|---|---|---|---|---|
| **Abundance** (log$_{10}$-transformed) | | | | | |
| Constant | 15.2998 | 1 | 15.2998 | 74.8265 | 0.0000 |
| Site class | 2.8096 | 2 | 1.4048 | 6.8704 | 0.0023 |
| Error | 10.2236 | 50 | 0.2045 | | |
| | | | | | |
| **Biomass** (log$_{10}$-transformed) | | | | | |
| Constant | 0.15174 | 1 | 0.15174 | 5.44012 | 0.0238 |
| Site class | 0.01202 | 2 | 0.00601 | 0.21546 | 0.8069 |
| Error | 1.39466 | 50 | 0.02789 | | |
| | | | | | |
| **Species density** | | | | | |
| Constant | 221.763 | 1 | 221.763 | 67.0728 | 0.0000 |
| Site class | 38.232 | 2 | 19.116 | 5.7817 | 0.0055 |
| Error | 165.315 | 50 | 3.306 | | |
| | | | | | |
| **Percentage mud** | | | | | |
| Constant | 2.86595 | 1 | 2.86595 | 75.1721 | 0.0000 |
| Site class | 1.72991 | 2 | 0.86495 | 22.6872 | 0.0000 |
| Error | 1.86813 | 49 | 0.03813 | | |

*Table 2: Abundance variations of the top 10 species over site classes, with ANOVA significance level and significance level of linear regression of abundance with median grain size.*

| **Species** | **Abundance m$^{-2}$ in site classes** | | | **ANOVA over site classes** | **Regression with median grain size** |
|---|---|---|---|---|---|
| | ambient | Recently dredged | recovery | p | p |
| *Lagis koreni* | 1 | 12 | 100 | 0,0049 | 0,0428 |
| *Pisione remota* | 584 | 0 | 30 | 0,0024 | 0,0000 |
| *Aonides paucibranchiata* | 536 | 0 | 20 | 0,0064 | 0,0000 |
| *Polygordius appendiculatus* | 228 | 0 | 15 | 0,0391 | 0,0000 |
| *Goniadella bobrezkii* | 76 | 12 | 25 | 0,2580 | 0,0001 |
| *Magelona johnstoni* | 92 | 12 | 5 | 0,0114 | 0,1102 |
| *Nephtys cirrosa* | 40 | 0 | 65 | 0,0467 | 0,1764 |
| *Notomastus latericeus* | 0 | 75 | 35 | 0,2360 | 0,8712 |

| *Kurtiella bidentata* | 4 | 0 | 55 | 0,1169 | 0,5855 |
| *Ophiura ophiura* | 4 | 0 | 45 | 0,1131 | 0,3317 |


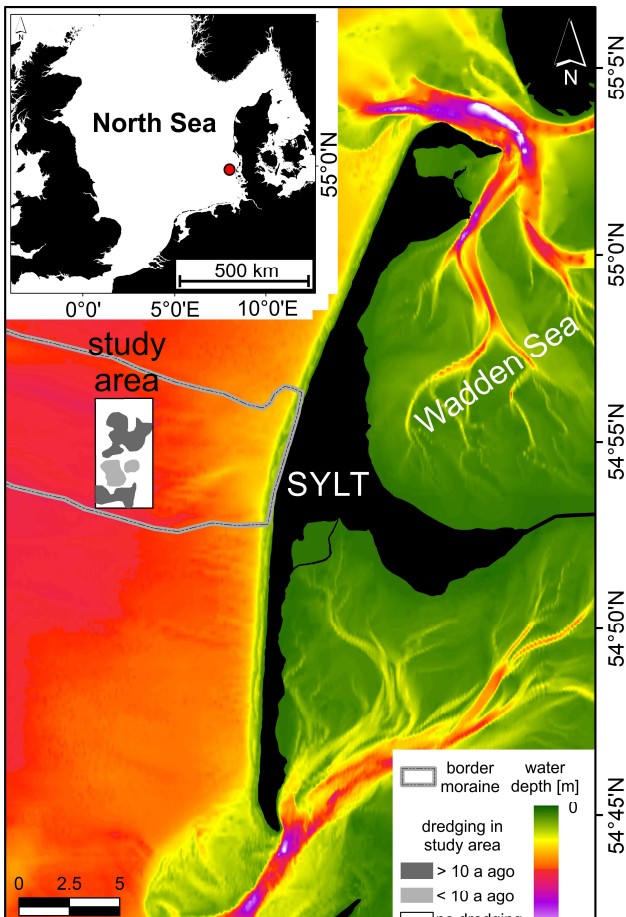

**Figure 1: Study area "Westerland dredging area" (WDA) located west of the Island of Sylt (SE North Sea). Bathymetric information were provided by the German Federal Maritime and Hydrographic Agency (BSH, 2018) and own measurements. Geological data were modified after Streif and Köster, 1978 (subaquatic border of the Saalian PISA-moraine).**

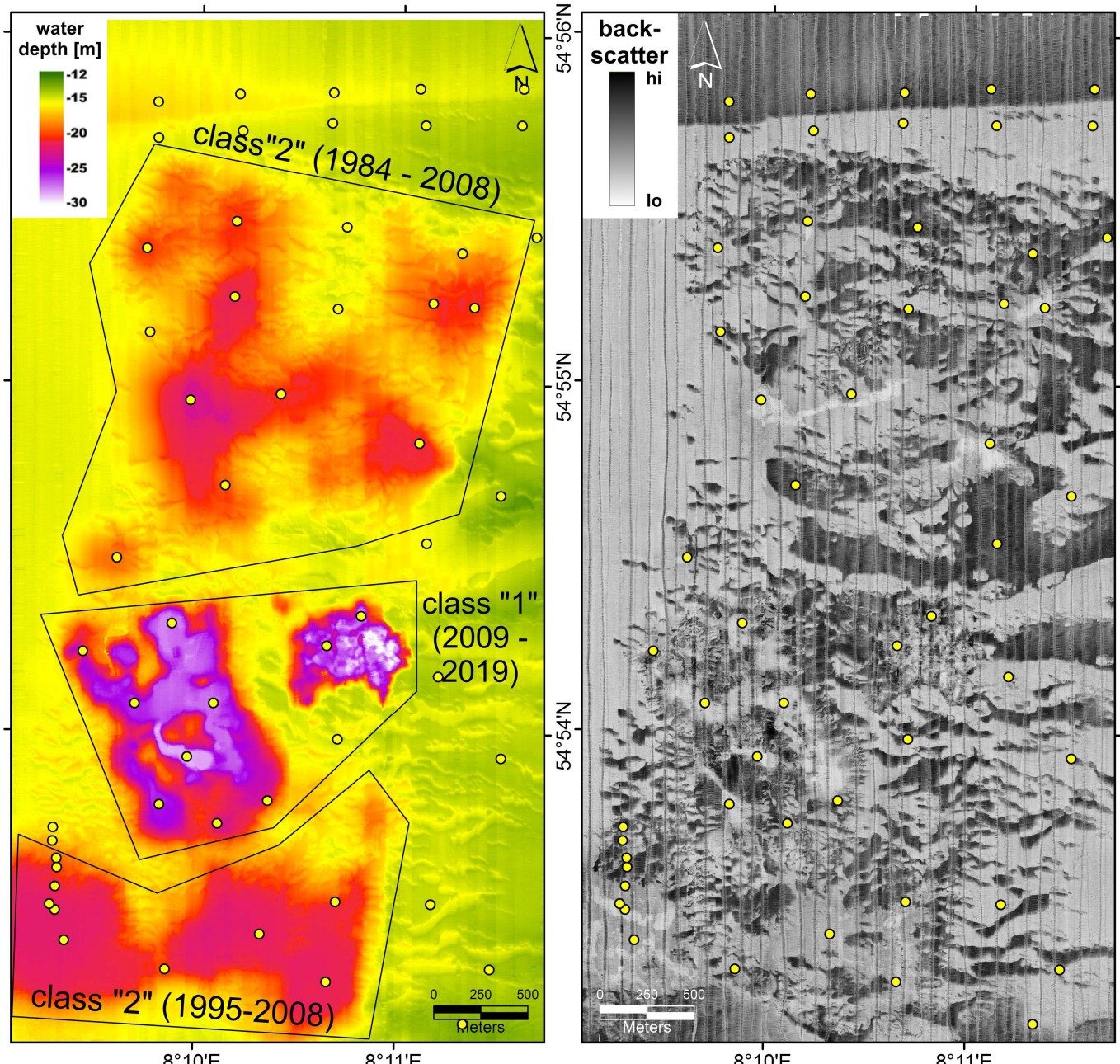

**Figure 2: Results of the hydroacoustic survey along 55 N-S transects executed in January 2019. Left: post-processed bathymetric map of the study site measured with multibeam echosounder; class "1": sites where sediment was extracted during the past 10 year (1984 – 2008); class "2": sites where sediment extraction terminated at least 10 years prior to the sampling (2009 – 2019); class "0": control sites which are directly unaffected by dredging (area outside the boxes). Right: Backscatter response of the seafloor recorded with sidescan sonar (here: 330 kHz); dark grey = high backscatter, light grey = low backscatter. Surveyed transects becomes visible as longish dark grey stripe in the sidescan mosaic proceeding in N-S direction. Position of grab sample stations are indicated by yellow dots.**

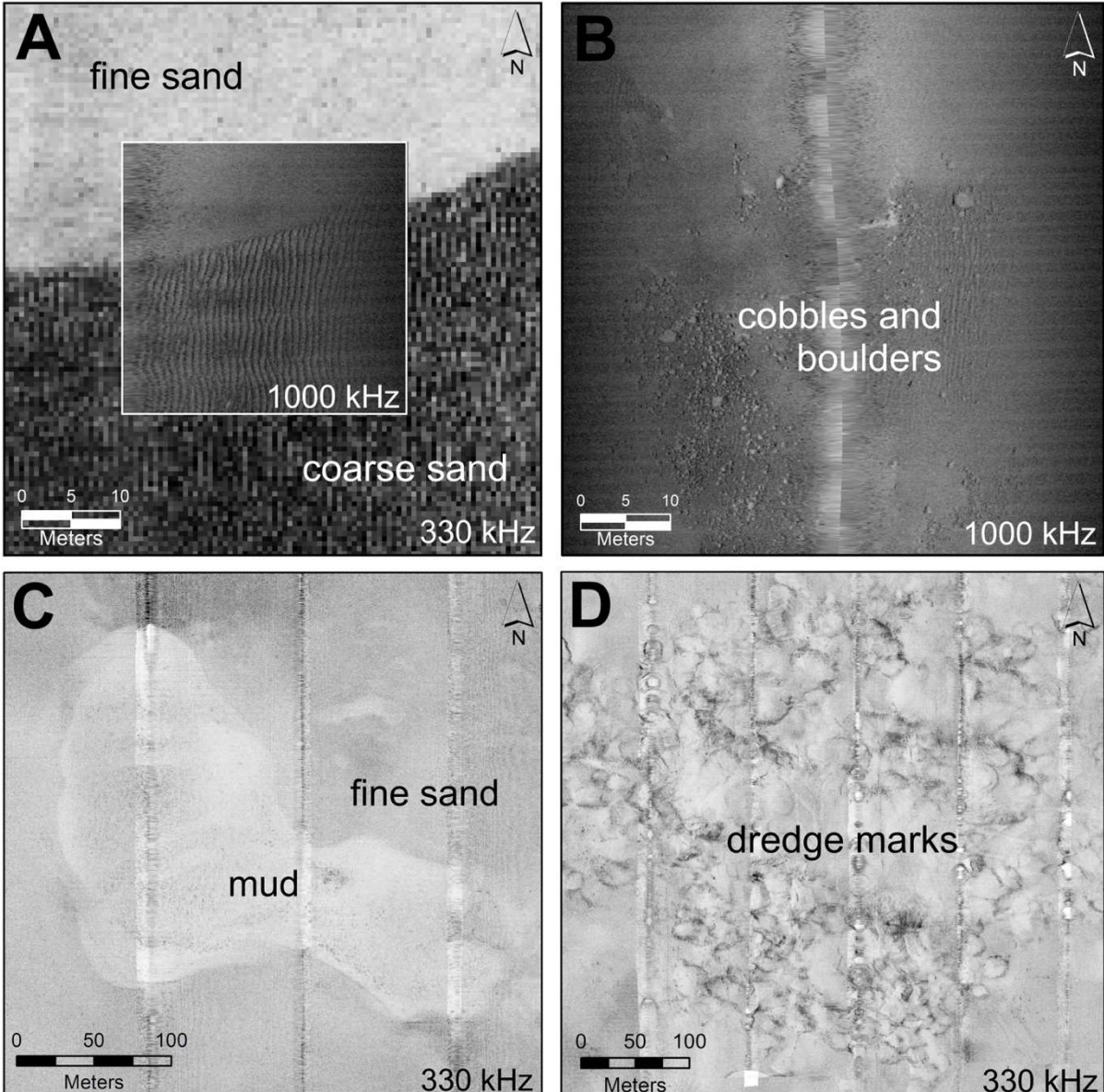

Figure 3: Seafloor features detected within the two sidescan sonar mosaics: (A) rippled coarse sand (dark) and smooth fine sand (bright) demarcated by a sharp border. (B) cobbles and boulders. In the direction of the sonar source, these objects initially exhibit high backscatter followed by a bright acoustic shadow from which no backscatter can occur. (C) very smooth mud area with low backscatter surrounded by a domain of fine sand with intermediate backscatter. (D) cone shaped funnels representing the dredging marks on the seafloor. Since higher backscatter values occur on the slopes of the funnels that are directed towards the sonar source differences in sediment distribution are not distinguishable using hydroacoustic.

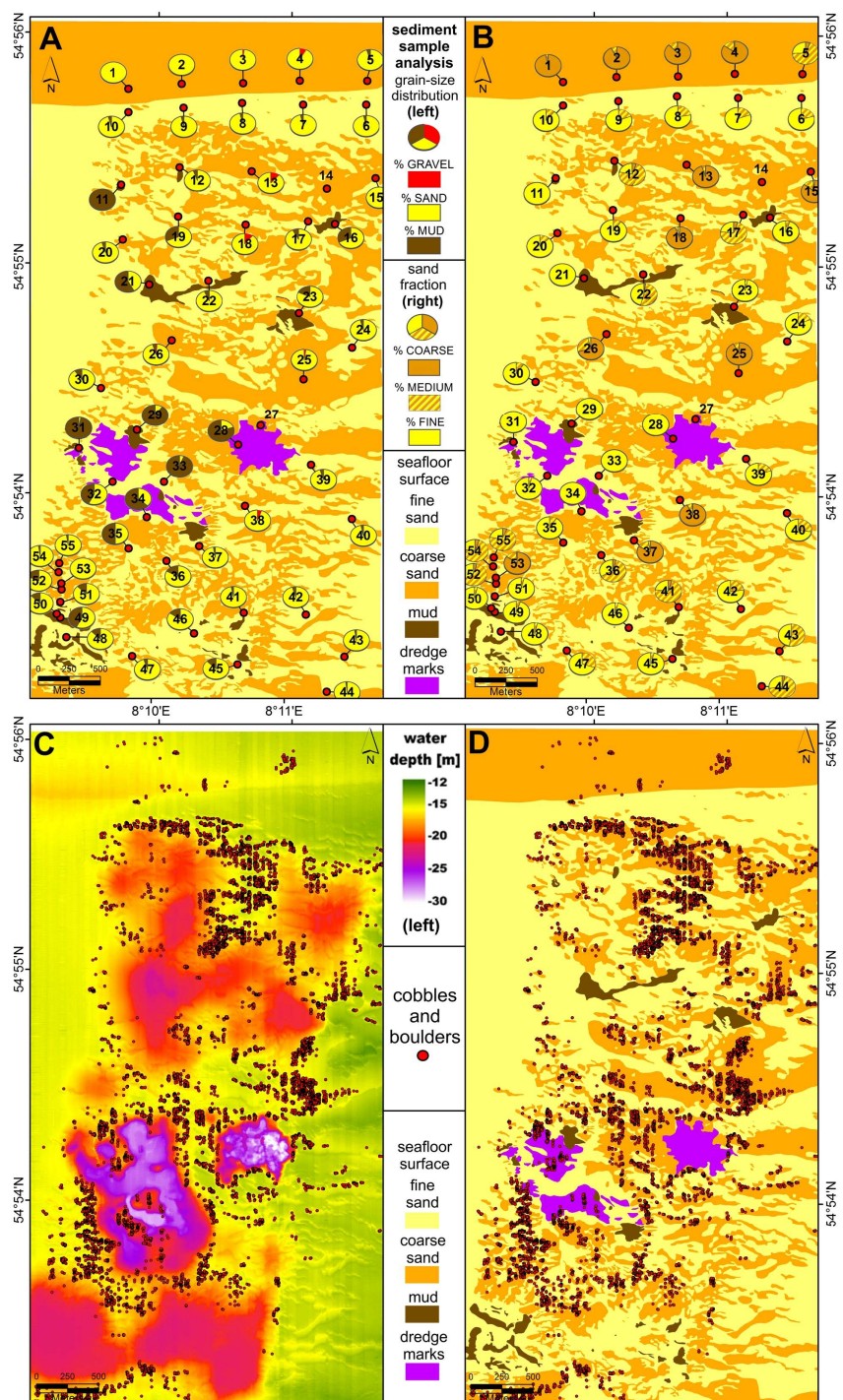

**Figure 4: Sediment distribution and bathymetric maps created with a combination of hydroacoustic data and ground-truth information. (a), (b): Position (red dots) and sediment composition of the grab samples. (c): appearance of stones compared to bathymetry. (d): appearance of stones compared to sediment distribution. For age of the dredging zones cf. Fig. 2 (left).**

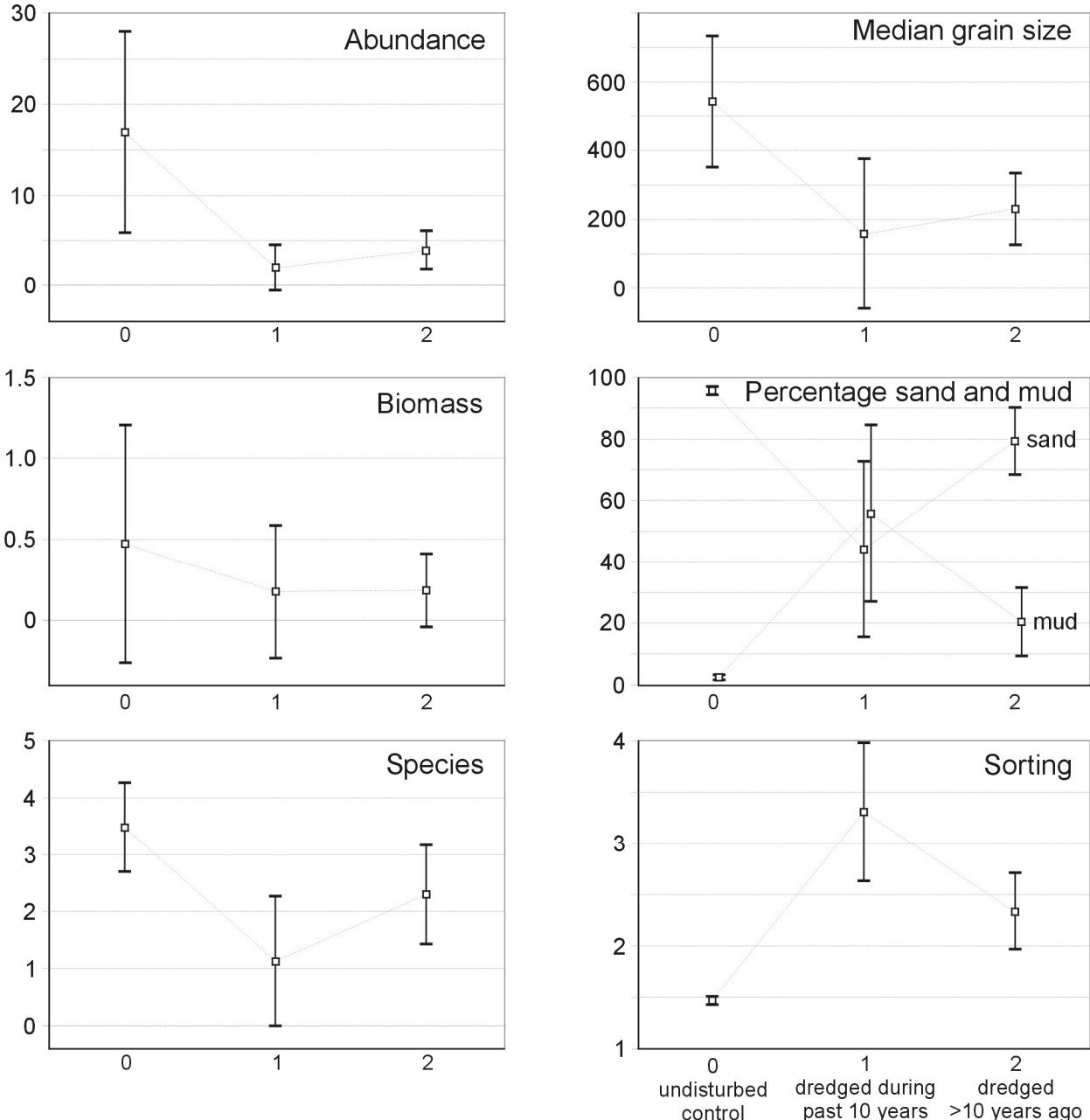

**Figure 5: Macrozoobenthos abundance, biomass and species density per sediment core and sediment properties of sampling stations across the sediment extraction area; means and 95% confidence intervals, biomass in g fresh weight. Site class 0 = control sites unaffected by sediment dredging; class 1 = sites dredged within the last 10 years; class 2 = sites >10 years after dredging.**

580