# Peer review of "Persistent effects of sand extraction on habitats and associated benthic communities in the German Bight"

_Biogeosciences, 2020_

## Referee Comment (RC1) · Anonymous Referee #1 · 12 May 2020

**Review of manuscript bg-2020-17: "Persistent effects of sand extraction on habitats and associated benthic communities in the German Bight"**

**General comments:**

This paper intends to investigate the effects of historic and recent intensive dredging on habitats and benthic fauna in the German Bight in a dredging area near Sylt. This is a follow-up paper of Mielck et al 2018 where the focus was on morphological changes due to sand extraction for beach nourishment. This definitely is of scientific value and interest in the field of effect studies and increasing demand of sand for both industrial purposes and coastal protection.

However, I had to find out myself that this was a 'follow-up' study and really needed to read the Mielck paper to get better insights in this study and understand the situation of the area. At least, this could have been better referred to. Furthermore, the way it is written and presented now, especially the discussion adds little value compared to the previous study. Although, in itself, it really is a different study and could add interesting new scientific insights. But therefore, this manuscript has to be thoroughly reworked with focus on the new aspects i.e. defining the different habitats related to the dredging history of the sites and characterizing the benthic communities related to these habitats. The introduction should therefore at least make a clear referral to the previous study and the conclusions of that one. Moreover the manuscript should better introduce the available knowledge on the topic of impact of sand extraction on benthic habitats since too few references have been cited, while already quite some literature is available and this would situate the study in a broader perspective. Objectives should also be better delineated to make clear what the main aim of this exact study was. This could also help maybe to explain the unconventional way of benthic sampling i.e. very small sample volume used for species identification compared to volume used for sediment analyses.

Results are too vague and too descriptive. Extra multivariate analyses should be done to characterize communities. Maybe acoustic data together with sediment data could be used in a PCA and these PCA results (=axis scores) can in their turn be used in the faunal analyses so that acoustic data are really used to determine benthic communities? This study would really benefit from a better combination of both datasets, since this is its strength. While in the current version of the manuscript, these two datasets are treated as separate entities.

Discussion is too superficial and adds very few new insights compared to the previous paper as well as said above. Plus it thus not really discuss the results of this study. I also do not agree with the conclusion made. The historic dredging actually caused a loss of habitat in my opinion. You even get a change in EUNIS habitat, so regeneration to the original habitat, without human intervention, will not be possible in that sense you cannot speak about regeneration/recovery. This could be discussed in the light of the MSFD, D6 seafloor integrity C1 habitat loss. See also specific comments for some extra input in the discussion that could lift it up to an interesting contribution for the scientific community. Also here, quite some literature is already available to put your results in a wider perspective but only very few references have been used. Looking into the existing body of literature and putting your results in this wider perspective would give more body to the discussion.

To conclude, the manuscript cannot be published in the current version, thorough revision is needed of all sections and some new analyses need to be done to make this a valuable contribution.

**Specific comments in chronological order:**

**Introduction**

L30-31: 'current' with references from 2010 is somewhat outdated in my opinion. I would suggest to check ICES WGEXT reports where recent figures are yearly reported for NE Atlantic countries. Latest report has figures from 2018 even making a distinction between extraction for coastal protection and for industrial purposes, see https://www.ices.dk/community/groups/Pages/WGEXT.aspx

L30-42: very few references while quite some papers have been published on these topics so would be good to support these lines with extra references. Just naming a few: Le Bot et al 2010, Foden et al 2009, Kubicki et al 2007, Van Lancker et al 2015, also in cooperative research report of ICES WG on extraction a lot of references are incorporated in chapter on ecological impact of sand extraction (http://ices.dk/sites/pub/Publication%20Reports/Cooperative%20Research%20Report%20(CRR)/CRR330.pdf)

L64-66: refine/rephrase your objectives – maybe better in the form of hypotheses, research questions?

L70 and further: this part should be moved to acknowledgements

**Study area**

L74: make 'study area' a section under M&M

L75 - Fig1: Please include location of reference area(s) plus add information (best in a more detailed zoom) on e.g. geological layers, bathymetry and past and 'recent' dredged areas on the map, so it is more in line with the information provided in study area paragraph

L78: water depths between 14 and 30 m, is this natural depth range or does this include extraction pits already? Confusing, I would suggest to report 'natural' depth

L79: would be good to add cumulative amount of sand that has been extracted since 1984

L82: what type of dredging is done? Static or trailer dredging?

L83: typo add IS derived

L84: 'prevails' strange wording, better 'takes place'?

**Material and methods**

L94 'all-over' replace by 'over-all', 'high-resolute' replace by 'high resolution'

L95-96: Please put transects and location of grab samples on a map.

L95: These 55 transects were done for both multibeam and sidescan? Simultaneously or on different days? Please provide information on this in M&M section. Also not clear how long survey was, all in one week, several days throughout January? January can be quite heavy weather and shallow area so weather can have influence on measurements, certainly when spread over several days. Info needed.

L111: please add what focus is of 330 kHZ and what of 1MHz sonar

L126: very unconventional way of sampling benthos, very small samples for macrobenthos…I would expect the other way around big enough subsample for sediment and main sample for benthos? Why was this done this way? Clear justification is needed

L129-130: class 0 control, is this really control, undisturbed conditions?? What about indirect/secondary impacts? Can you be certain that these are not at all affected by the dredging?

**Results**

L140: replace 'excavation' by 'extraction' – try to be consistent throughout the manuscript

L141: how do you know they are only partially refilled? What was depth after cessation of extraction? What is depth now? Please support your statements with numeric data.

L141-145: in text, you mention letters a, b, c , d but these are not indicated on figure2. Please make sure that your figures and text match. – actually these are results from a previous study so delete here? Or clarify, since now it is confusing because you refer to/compare with earlier published study and description of these dredging pits is not the aim of this study.

L146-147: please define what is high, intermediate and low backscatter

L156-157 – Fig4: would be good to have delineation of different dredging zones cf. old, new ones this would make interpretation of the maps more clear.

L162: on which results this statement is based? Data in results are needed to support this? E.g. multivariate analyses or cluster analyses.

L163-165: ? would think this is a result that should be under habitat mapping? This is ground truthing of your sidescan results

L166-170: unclear – is there a difference between undisturbed and control? First time mentioned in the paper. Please rephrase.

L174-177: idem as comment above, please use multivariate analyses to back up these statements with SIMPER to demonstrate which species are making the difference between the groups

**Discussion**

L190: okay, low sedimentation rates but how does the mud comes in? Do these pits not act as traps for mud?

L191-193: this is for the first time you mention the earlier study with which you compare the 2019 measurements – this causes the reader to be very confused all the way throughout the paper. Should be made clear from the beginning, even in introduction results of previous study should be situated.

L212: is this a successional state? In my opinion, this is just an altered habitat which will never recover to the old state and reach a different equilibrium or has already reached it in the historic dredging pits. I would call this physical loss of benthic habitat (cf. MSFD descriptor 6 C1) due to dredging, even at the EUNIS level. For more background information see reports on this topic http://www.ices.dk/sites/pub/Publication%20Reports/Expert%20Group%20Report/Fisheries%20Resources%20Steering%20Group/2019/WKBEDPRES2/WKBEDPRES2_Report_2019.pdf + related info. Very weak discussion regarding the benthic results/benthic habitats. This is focus of the paper but it fails to add body to this topic while it actually should be the main part of the discussion. Combining the acoustic data with the biological data, is the interest of this paper.

L213-214: what are these habitat types? Where do you find which one? How related to dredging history? This is what should be discussed? Which habitat type, you found where and what are the indicator species for this type of habitat? As said above, this should be focus of discussion.

L216: do you want restoration? Naturally, it will probably not be possible? So if you want restoration maybe mitigation through human intervention is needed? Or if not, leave it like it is, other suggestions? What would be frequency needed for monitoring, yearly, every 5 years? Every 10

years? As you suggest, rate of infilling is very slow so why put money in a monitoring study where you already know the result? Wouldn't it be better to put the money in other research questions or mitigation measures or 'working with nature' designs? Or studies to prevent this happening again with the ongoing dredging? I am just putting forward some ideas, topics that could be included in the discussion and that would give it more scientific value. Now the discussion is too superficial, it could be lifted up by going more into depth in the main topics of your study.

---

## Referee Comment (RC2) · Anonymous Referee #2 · 24 May 2020

Review of manuscript bg-2020-17: "Persistent effects of sand extraction on habitats and associated benthic communities in the German Bight"

General comments:

This paper investigates the recovery of benthic invertebrate fauna following dredging for beach replenishment in the German Bight. The under lying premise of this work is that eventually the habitat condition will return to pre-dredging condition, and this will allow the re-establishment of pre-dredging benthic fauna, which is found in nearby undisturbed areas. The author suggests that any benthic assemblage that differs from the unimpacted zones is a successional stage.

This premise has 2 flaws:

1. The author states that the sands being dredged are Pleistocene in age, and that the existing hydraulics of the system and sediment supply limits deposition to very fine sand and mud at very low rates of deposition. This information highlights that the hydraulics and sediment supply of the area has fundamentally changed since the sands were deposited in the Pleistocene, so it is difficult to understand how the author expects these conditions to be restored, even over many decades/centuries/millenia. In effect, Pleistocene marine conditions and sediment supply would need to be re-established for the pre-mining conditions to be re-sustablished.  What these changes do highlight is that the dredging is an unsustainable mining activity – e.g. the material being extracted will not be replenished, so the dredging is resulting in a permanent change in habitat conditions. The implications of this permanent change should be the focus of the paper.
2. The benthic assemblages present at the mined sites are not a successional stage that will ultimately lead back to the pre-mining assemblage. If the muds are not replaced by sand (which is highly unlikely given the quiescent hydraulics and limited sediment supply), then the mud loving assemblage will remain in perpetuity. The paper suggests that there is a successional order of benthic infauna, while at the same time saying the infauna reflects the sediment characteristics. Mud is not a successional stage to sand. The change in benthos due to the change in sediment, and why this matters, should be the focus of the discussion, not that it is an intermediate step leading back to the original assemblage.

This work should be re-framed to highlight the permanent changes that are occurring to the sediments, how the infauna has changed due to these impacts, and what are the implications of these changes. The author states that the sandy benthos is wide-spread and the mining is not a threat to the prevailing species – so the question is what are the implications, if any, of the conversion of sandy habitat to muddy habitat and the loss of sessile habitat? Discussing how these changes might affect other trophic levels or food webs, such as through the uptake of PAHs or other contaminants from the mud, would be more relevant than focussing on the (lack of) re-establishment of the original fauna. It would also provide more context for the comment about monitoring PAH's which is otherwise unrelated to anything discussed in the paper.

The introduction and presentation of the scientific question should be revised and strengthened, and the aim of the investigation should be clearly stated earlier in the paper. It would also be useful to provide more discussion about why deeper extraction pits might have a different recolonization trajectory as compared to shallower disturbances. Providing some hypothetical examples of how deeper disturbance could have different impacts as compared to shallow extraction would provide more context for the results. The paper would benefit from additional, and more recent references.

Other comments:

1. First sentence in abstract is very ambiguous – what local population is being referred to? What is sea-level rise demanding?
2. This paper discusses regeneration whereas it might be more applicable to use re-establishment.
3. Line 29 – has reached a high level? Examples of the growth in extraction rates would be useful if it is considered that ongoing sand mining will pose a threat.
4. Line 36 – activities have led
5. Line 54 - sonars allow the analysis of backscatter intensity – how? This needs more explanation.
6. Line 59 – poor English
7. Line 64 introduces aim of paper – should be presented much earlier
8. Line 67 -will be used? Have been used.
9. 53 grab samples for 55 x 5 km transects are not a lot of samples. A more detailed description of the sampling strategy should be presented to demonstrate the samples are representative of the different areas.
10. Line 99 - area was collected
11. Some justification should be provided that all of the past mining pits are still visible. Were the locations compared to maps? How would you know if a pit was no longer visible?
12. Benthos -line 159 – what is class 1 lower than? Class 0?
13. Polychaete profited? Polychaete exploited suitable environment.

---

## Author Comment (AC1) · 11 Jun 2020

**Review of manuscript bg-2020-17: "Persistent effects of sand extraction on habitats and associated benthic communities in the German Bight"**

**General comments:**

This paper intends to investigate the effects of historic and recent intensive dredging on habitats and benthic fauna in the German Bight in a dredging area near Sylt. This is a follow-up paper of Mielck et al 2018 where the focus was on morphological changes due to sand extraction for beach nourishment. This definitely is of scientific value and interest in the field of effect studies and increasing demand of sand for both industrial purposes and coastal protection. However, I had to find out myself that this was a 'follow-up' study and really needed to read the Mielck paper to get better insights in this study and understand the situation of the area. At least, this could have been better referred to.

Furthermore, the way it is written and presented now, especially the discussion adds little value compared to the previous study. Although, in itself, it really is a different study and could add interesting new scientific insights. But therefore, this manuscript has to be thoroughly reworked with focus on the new aspects i.e. defining the different habitats related to the dredging history of the sites and characterizing the benthic communities related to these habitats. The introduction should therefore at least make a clear referral to the previous study and the conclusions of that one. Moreover the manuscript should better introduce the available knowledge on the topic of impact of sand extraction on benthic habitats since too few references have been cited, while already quite some literature is available and this would situate the study in a broader perspective.Objectives should also be better delineated to make clear what the main aim of this exact study was. This could also help maybe to explain the unconventional way of benthic sampling i.e. very small sample volume used for species identification compared to volume used for sediment analyses. Results are too vague and too descriptive. Extra multivariate analyses should be done to characterize communities. Maybe acoustic data together with sediment data could be used in a PCA and these PCA results (=axis scores) can in their turn be used in the faunal analyses so that acoustic data are really used to determine benthic communities? This study would really benefit from a better combination of both datasets, since this is its strength. While in the current version of the manuscript, these two datasets are treated as separate entities. Discussion is too superficial and adds very few new insights compared to the previous paper as well as said above. Plus it thus not really discuss the results of this study. I also do not agree with the conclusion made. The historic dredging actually caused a loss of habitat in my opinion. You even get a change in EUNIS habitat, so regeneration to the original habitat, without human intervention, will not be possible in that sense you cannot speak about regeneration/recovery. This could be discussed in the light of the MSFD, D6 seafloor integrity C1 habitat loss. See also specific comments for some extra input in the discussion that could lift it up to an interesting contribution for the scientific community. Also here, quite some literature is already available to put your results in a wider perspective but only very few references have been used. Looking into the existing body of literature and putting your results in this wider perspective would give more body to the discussion. To conclude, the manuscript cannot be

published in the current version, thorough revision is needed of all sections and some new analyses need to be done to make this a valuable contribution.

*Dear Reviewer #1,*

*Thank you for your revision and the numerous helpful comments and suggestions for improvement. Indeed, this manuscript is something like a follow-up to one of our previous study (Mielck et al. 2018). The focus of that study was set on morphological changes due to marine aggregate extraction in the same study area using bathymetric data between 1993 and 2017. For the new study presented here, we collected new data and intended to focus on the impact of sand extraction on the habitats and the associated benthic communities. You are right that the conclusion of the previous work needs to be better communicated in the introduction or at least in the section "Study area". We will certainly address this weakness.*

*Many thanks also for the provided literature and the hint to the ICES WGEXT reports, which are very helpful to improve our introduction with more facts and recent information about this topic, which is also useful for deeper discussion and a meaningful conclusion.*

*We think that a better delineation of the aims of this study towards a combination of the used data sets (benthos analysis and hydroacoustic data) is a very good way to improve the whole study. We already started some statistical analysis (e.g. SIMPER) and think that this will add more insights into the habitat characteristics.*

*A recovery towards the pre-dredging conditions is of course not possible since the coarse Pleistocene sediment cannot be replaced without a new ice-age. When sand mining started in 1984, the coastal authorities and also some scientist assumed, the so-called "Wanderfeinsand" (migrating fine sand, Tabat, 1979; Köster 1979), which is ubiquitous in the German Bight, will refill the pits relatively quick. This did not happened until now because of to weak sedimentation rates. However, a recovery to a fine sand habitat might be possible (maybe in decades or centuries). When the pits are flattened enough also current velocity will increase again. This would prevent an accumulation of muddy material.*

The word "regeneration" will adequately been substituted with "recovery" and "re-establishment" throughout the whole manuscript.

**Specific comments in chronological order:**

Introduction

L30-31: 'current' with references from 2010 is somewhat outdated in my opinion. I would suggest to check ICES WGEXT reports where recent figures are yearly reported for NE Atlantic countries. Latest report has figures from 2018 even making a distinction between extraction for coastal protection and for industrial purposes, see https://www.ices.dk/community/groups/Pages/WGEXT.aspx

*Thank you for the information about the latest data.*

*While between 1998 and 2002 approximately 53 million m³ was extracted, a total of 73.2 million m³ was extracted from the northern European Continental Shelf in 2018 (ICES, 2016, ICES, 2019).*

L30-42: very few references while quite some papers have been published on these topics so would be good to support these lines with extra references. Just naming a few: Le Bot et al 2010, Foden et al 2009, Kubicki et al 2007, Van Lancker et al 2015, also in cooperative research report of ICES WG on extraction a lot of references are incorporated in chapter on ecological impact of sand extraction (http://ices.dk/sites/pub/Publication%20Reports/Cooperative%20Research%20Report%20(CRR)/CRR 330.pdf)

*The quoted reference are very helpful and a good support. Additionally they are a good basis for further literature. Thank you.*

L64-66: refine/rephrase your objectives – maybe better in the form of hypotheses, research questions?

*As already mentioned above, we think, that the aims and objectives should be adjusted towards new statistical analysis. Maybe like this:*

*"The aim of this study was to further determine the impacts of extensive marine aggregate extraction on the regional macrozoobenthic communities. The main objectives were to (i) gain a deeper understanding of the correlation between the prevailing habitats and the recovery state of the associated benthic assemblages, to (ii) evaluate temporal recovery patterns along with short- and long-term changes in the community structures and to (iii) investigate the potential of a re-establishment of pre-dredging conditions regarding fine sand domains (Pleistocene material cannot be re-established). Therefore, dredging pits of different ages and, as a control, the sandy areas surrounding the extraction site were compared for sediment and benthic faunal composition. Using hydroacoustic gear and sediment grab samples, habitat maps were created combining sediment properties with information about abundance and diversity of the macrozoobenthos."*

L70 and further: this part should be moved to acknowledgements

*Indeed.*

**Study area**

L74: make 'study area' a section under M&M

*Good Idea.*

L75 - Fig1: Please include location of reference area(s) plus add information (best in a more detailed zoom) on e.g. geological layers, bathymetry and past and 'recent' dredged areas on the map, so it is more in line with the information provided in study area paragraph

*We already revised Fig. 1 and added bathymetric information as well as the alignment of the Saalian moraine, from which the sediment has been extracted. Additionally, we include the locations of differently-aged extraction areas and the reference area. The zoom-factor was also increased.*

[Figure]

**Figure 1: Study area "Westerland dredging area" located west of the Island of Sylt (SE North Sea). Bathymetric information were provided by the German Federal Maritime and Hydrographic Agency (BSH, 2018) and own measurements. Geological data were modified after Streif and Köster, 1978 (subaquatic border of the Saalian PISA-moraine).**

L78: water depths between 14 and 30 m, is this natural depth range or does this include extraction pits already? Confusing, I would suggest to report 'natural' depth

*The stated water depths include the extraction pits. The seafloor in the study area and also in the surroundings is very flat. The natural water depth ranges between 14 and 17.5 m in the research site.*

L79: would be good to add cumulative amount of sand that has been extracted since 1984

*Since 1984, more ~41 Mio m³ were extracted.*

L82: what type of dredging is done? Static or trailer dredging?

*Dredging was achieved using Trailing suction Hopper Dredgers.*

L83: typo add IS derived

*Ok. Thank you*

L84: 'prevails' strange wording, better 'takes place'?

*Yes, of course.*

**Material and methods**

L94 'all-over' replace by 'over-all', 'high-resolute' replace by 'high resolution'

*Indeed*

L95-96: Please put transects and location of grab samples on a map.

*Transects and positions of the grab samples are provided in the section 'results'. We can add this information to the manuscript at this section. (Captions of Fig. 2 and Fig. 4a can be modified to highlight this information.)*

L95: These 55 transects were done for both multibeam and sidescan? Simultaneously or on different days? Please provide information on this in M&M section. Also not clear how long survey was, all in one week, several days throughout January? January can be quite heavy weather and shallow area so weather can have influence on measurements, certainly when spread over several days. Info needed.

*During the survey, which took place between January 25$^{th}$ and January 27$^{th}$ at calm weather conditions, multibeam echosounder and sidescan sonars were used simultaneously on all transects. Ground truthing comprised 53 grab samples for grain-size analysis and macrobenthic fauna and were done on January 31$^{th}$.*

L111: please add what focus is of 330 kHZ and what of 1MHz sonar

*Using different frequencies result in more detailed information about the seafloor environment. Sidescan mosaics recorded with a low frequency generally yield information about large-scale objects on the seafloor (e.g. facies changes, sandwaves, megaripples) while a high frequency give more information about small-scale structures (e.g. ripple marks, stones (Mielck et al. 2015)).*

L126: very unconventional way of sampling benthos, very small samples for macrobenthos…I would expect the other way around big enough subsample for sediment and main sample for benthos? Why was this done this way? Clear justification is needed

*Initially, this study was planned as a follow-up to investigate the further sedimentation characteristics of the dredging holes. When deep sediment dredging was first applied, the deeper water layer and sediment inside the dredging holes temporarily depleted in oxygen and became close to azoic (Armonies & Buschbaum 2008, unpublished report for the national authorities). This was thought to be due to the diameter/depth aspect favouring stagnant conditions in relatively small but deep holes. As a consequence, the national authorities decided that further dredging should use the same holes, i.e. increase their diameter instead of creating many small holes. The present study was an opportunity to check whether or not the larger diameter of the dredging holes would faciliate water circulation and thus enable permanent establishment of a macrozoobenthic community in the depths of the holes. Accordingly, the main questions to answer were*

*(1) is there life macrozoobenthos in the dredging holes?*

*(2) if yes, typical sediment-specific (i.e. mud-dwelling) fauna or just temporary opportunists washed into the holes from ambient sediments?*

*(3) is the benthic biomass inside the holes already comparable to ambient sediments, i.e. is the function as a potential feeding ground for higher trophic levels already restored?*

*To answer these simple questions, small sediment cores taken along with the sediment ground-truthing samples were considered adequate. A full description of the benthic communities in- and outside the dredging holes was not intended in this study. Only after the present results, it is clear that the current state of macrozoobenthic development warrants further studies with focus on the benthic communities, and therefore, with a sampling design adequate to reveal far more details of benthic community composition.*

*Armonies, W. , Buschbaum, C. (2008): Fachgutachten Makrozoobenthos im Rahmen der UVS für das Sandentnahmegebiet "Westerland III" westlich von Sylt. Im Auftrag des Landesbetrieb für Küstenschutz, Nationalpark und Meeresschutz Schleswig-Holstein, pp. 1-94.*

L129-130: class 0 control, is this really control, undisturbed conditions?? What about indirect/secondary impacts? Can you be certain that these are not at all affected by the dredging?

*We know the seafloor west of Sylt very well. For example, in Mielck et al. 2015 we made a study in an area ca. 6 km northeast of the dredging area. Additionally, during the joint research project WIMO (2010-2015), we investigated several study areas more than 20 km south of it. The investigation shows that the seafloor conditions are very similar to the conditions in class "0" (patterns of fine and coarse sand -> so called sorted bedforms). During the dredging process in our study area, only small portions of very fine material are released to the water. This fraction was not detected in the grain-*

*size spectrum of our sediment samples in this area west of Sylt. It seems that the material is transported away by tidal currents in northern direction.*

*A direct impact due to dredging activities in class "0" can be excluded, since this is not permitted by the coastal authorities. In addition, the bathymetry does not show any signs of dredging.*

**Results**

L140: replace 'excavation' by 'extraction' – try to be consistent throughout the manuscript

*No problem.*

L141: how do you know they are only partially refilled? What was depth after cessation of extraction? What is depth now? Please support your statements with numeric data.

*We showed the refill process in a previous paper about the dredging area (Mielck et al. 2018: Morphological changes due to marine aggregate extraction for beach nourishment in the German Bight (SE North Sea)). Here, we also used data from 1993, that showed water depth of ~26 m in the northern pit. At this point of time, dredging activities were still conducted in this part of the study area. In 2008, water depths were at ~22 m and 10 years later, no measureable differences in water depth occurred. Moreover, we made investigations with a seismic device that revealed old slope failures in the subbottom, which are an indicator for a refill process.*

*Additionally, data from Zeiler et al. (2004) reveal that similar dredging depths (~ 33 m water depth) were also achieved in the year 1991. The dredging depth after cessation seemed to be generally at this level in order to limit the size of impact.*

L141-145: in text, you mention letters a, b, c , d but these are not indicated on figure2. Please make sure that your figures and text match.

*Yes, this can be done very easily.*

– actually these are results from a previous study so delete here? Or clarify, since now it is confusing because you refer to/compare with earlier published study and description of these dredging pits is not the aim of this study.

*You are right, that the refill process is a result from the previous study. However, we think that it is important to show at least a bathymetric map of the study area and to give a hint to the refill process. Hence, we can add a reference to Mielck et al. 2018 at this place.*

*Note: The bathymetry shown here was recorded in 2019 and was not published in Mielck et al. 2018.*

L146-147: please define what is high, intermediate and low backscatter

*The backscatter of the seafloor is illustrated in a grey-scale. When taking a look on Fig. 2 (right) you can see, that there are three different backscatter classes in the sonar-mosaic consisting of a range of*

*grey values(high, intermediate, low backscatter). They represent three domains: coarse sand, fine sand, and mud. Since the backscatter in the hydroacoustic is a complex process, it is not easy to define exact borders at the 8-bit color palette. There were many attempts in the past; however, the backscatter could be affected by very many parameters: e.g. the slope of the seafloor, the distance between the seafloor and the transducer, the slant range and angle of inclination, the gain-settings etc.*

*The post-processing software SonarWiz, which we used, has an option for automatic classification. However, there are always some artefacts in the resulting maps. Hence, we prefer to show the "raw" data in a sonar mosaic and do a supervised classification of the habitats with ARCGIS aided by our ground-truth data.*

*We can change this phrase in the manuscript, in order to clarify this aspect.*

L156-157 – Fig4: would be good to have delineation of different dredging zones cf. old, new ones this would make interpretation of the maps more clear.

*These delineations are already included in Figure 2 (left). Fig. 4 already yields many features and we think that it is not a good idea to add more lines to the figure. However, we can put a hint in the caption regarding the ages: e.g.: "For age of the dredging zones cf. Fig. 2 (left)."*

L162: on which results this statement is based? Data in results are needed to support this? E.g. multivariate analyses or cluster analyses.

*As already mentioned: We will provide additional SEMPER and ANOSIM statistics/results.*

L166-170: unclear – is there a difference between undisturbed and control? First time mentioned in the paper. Please rephrase.

*No, there is no difference and this will be rephrased.*

L174-177: idem as comment above, please use multivariate analyses to back up these statements with SIMPER to demonstrate which species are making the difference between the groups

*Yes, we will do it (see above).*

**Discussion**

L190: okay, low sedimentation rates but how does the mud comes in? Do these pits not act as traps for mud?

*Yes, this is correct. They act as sediment traps. The current velocity decreases inside the pits and muddy material, which is in suspension (most likely coming from rivers) accumulates in the pits. This is one result from the previous study (Mielck et al. 2018) and was also a finding from Zeiler et al., 2004, who measured current velocities inside the dredging pits. This information should be added to the manuscript.*

*"The current velocity strongly decrease inside the pits which allows the suspended mud to accumulate on the seafloor (Zeiler et al., 2004)."*

L191-193: this is for the first time you mention the earlier study with which you compare the 2019 measurements – this causes the reader to be very confused all the way throughout the paper. Should be made clear from the beginning, even in introduction results of previous study should be situated.

*To set it in a better context, this information can be added to the section 2.1 Study area. E.g.: "This study is a follow-up to the previous study Mielck et al. 2018, which focused on morphological changes due to marine aggregate extraction in this study area using bathymetric data between 1993 and 2017."*

L212: is this a successional state? In my opinion, this is just an altered habitat which will never recover to the old state and reach a different equilibrium or has already reached it in the historic dredging pits. I would call this physical loss of benthic habitat (cf. MSFD descriptor 6 C1) due to dredging, even at the EUNIS level. For more background information see reports on this topic http://www.ices.dk/sites/pub/Publication%20Reports/Expert%20Group%20Report/Fisheries%20Resources%20Steering%20Group/2019/WKBEDPRES2/WKBEDPRES2_Report_2019.pdf + related info. Very weak discussion regarding the benthic results/benthic habitats. This is focus of the paper but it fails to add body to this topic while it actually should be the main part of the discussion. Combining the acoustic data with the biological data, is the interest of this paper.

*As already mentioned above: Maybe a recovery to a common fine sand habitat is possible although the coarse sand is lost forever. When the pits are flattened enough also current velocity will increase again. This would prevent an accumulation of muddy material. We will check your provided information, which surly will be a good basis for further discussion.*

L213-214: what are these habitat types? Where do you find which one? How related to dredging history? This is what should be discussed? Which habitat type, you found where and what are the indicator species for this type of habitat? As said above, this should be focus of discussion.

We will improve our discussion based on your suggestions.

L216: do you want restoration? Naturally, it will probably not be possible? So if you want restoration maybe mitigation through human intervention is needed? Or if not, leave it like it is, other suggestions?

*See above. We will think about it and consider to mentioning this in the discussion.*

What would be frequency needed for monitoring, yearly, every 5 years? Every 10

*During our investigations in the past years, we did a semiannual monitoring program in this area. The dredging season is generally between May and September. Before they started the extraction in spring 2017, we started our monitoring program to evaluate the situation and what happened to the new dredging pits from the last season during winter storms. When the dredging season was finished in September, we did a second survey to detect the new dredging pits and so on the next years. Since the analysis of the benthic communities is relatively great effort, we would do this only every two years in the future.*

*We could at this Information to the manuscript:*

*"As a strategy to monitor the further development in the extraction sites, we suggest semiannual investigations of the different habitat types by hydroacoustic means combined with the analysis of the benthic communities every two years."*

years? As you suggest, rate of infilling is very slow so why put money in a monitoring study where you already know the result? Wouldn't it be better to put the money in other research questions or mitigation measures or 'working with nature' designs? Or studies to prevent this happening again with the ongoing dredging? I am just putting forward some ideas, topics that could be included in the discussion and that would give it more scientific value. Now the discussion is too superficial, it could be lifted up by going more into depth in the main topics of your study.

*A regular monitoring is very meaningful, as the dredging activities will take place each year. We believe, hence, that these activities should be monitored by collecting hydroacoustic and benthic data on the same temporal scale. It will further allow us to control for other potential external influences, such as storm events. However, we will come up with ideas, how the impact of dredging might be reduced.*

*When we started the monitoring program, it was the first time that sophisticated multibeam devices were used in this area. We already knew that the accumulation rate is very slow. However, we did not know that it is that slow. Additionally, we had no idea, what happened to the benthic communities after more than 30 years after the impact.*

*However, aspects like working with nature or mitigation measures are important and should be mentioned in the discussion as prospection for the future.*

---

## Author Comment (AC2) · 11 Jun 2020

**Response to Reviewer #2**

Review of manuscript bg-2020-17: "Persistent effects of sand extraction on habitats and associated benthic communities in the German Bight"

General comments:
This paper investigates the recovery of benthic invertebrate fauna following dredging for beachreplenishment in the German Bight. The under lying premise of this work is that eventually the habitat condition will return to pre-dredging condition, and this will allow the re-establishment of predredging benthic fauna, which is found in nearby undisturbed areas. The author suggests that any benthic assemblage that differs from the unimpacted zones is a successional stage.

This premise has 2 flaws:
1. The author states that the sands being dredged are Pleistocene in age, and that the existing hydraulics of the system and sediment supply limits deposition to very fine sand and mud at very low rates of deposition. This information highlights that the hydraulics and sediment supply of the area has fundamentally changed since the sands were deposited in the Pleistocene, so it is difficult to understand how the author expects these conditions to be restored, even over many decades/centuries/millenia. In effect, Pleistocene marine conditions and sediment supply would need to be re-established for the pre-mining conditions to be re-established. What these changes do highlight is that the dredging is an unsustainable mining activity – e.g. the material being extracted will not be replenished, so the dredging is resulting in a permanent change in habitat conditions. The implications of this permanent change should be the focus of the paper.
2. The benthic assemblages present at the mined sites are not a successional stage that will ultimately lead back to the pre-mining assemblage. If the muds are not replaced by sand (which is highly unlikely given the quiescent hydraulics and limited sediment supply), then the mud loving assemblage will remain in perpetuity. The paper suggests that there is a successional order of benthic infauna, while at the same time saying the infauna reflects the sediment characteristics. Mud is not a successional stage to sand. The change in benthos due to the change in sediment, and why this matters, should be the focus of the discussion, not that it is an intermediate step leading back to the original assemblage.
This work should be re-framed to highlight the permanent changes that are occurring to the sediments, how the infauna has changed due to these impacts, and what are the implications of these changes. The author states that the sandy benthos is wide-spread and the mining is not a threat to the prevailing species – so the question is what are the implications, if any, of the conversion of sandy habitat to muddy habitat and the loss of sessile habitat? Discussing how these changes might affect other trophic levels or food webs, such as through the uptake of PAHs or other contaminants from the mud, would be more relevant than focusing on the (lack of) re-establishment of the original fauna.
It would also provide more context for the comment about monitoring PAH's which is otherwise unrelated to anything discussed in the paper.

The introduction and presentation of the scientific question should be revised and strengthened, and the aim of the investigation should be clearly stated earlier in the paper. It would also be useful to provide more discussion about why deeper extraction pits might have a different recolonization trajectory as compared to shallower disturbances. Providing some hypothetical examples of how deeper disturbance could have different impacts as compared to shallow extraction would provide more context for the results. The paper would benefit from additional, and more recent references.

Dear Reviewer #2,

Thank you for you revision and your helpful comments.

You are right. Investigations show, that something like a new ice age with strong glacification and interglacials are necessary to reach pre-dredging conditions. However, this knowledge is relatively new regarding this study area. When sand mining started in 1984, it was assumed, that the so-called "Wanderfeinsand" (freely translated: moving fine sand, Tabat, 1979; Köster 1979) would refill the pits relatively quick. Of course, the Pleistocene material is gone forever, but a refill with fine sand would not indicate such a strong change in habitat characteristics when compared to muddy domains, since fine sand domains are relatively common west of Sylt. When you have a look at Figure 4, you can see, that the fine sand superimposes the Pleistocene material quite regularly. Hence, also a complete refill with fine sand (maybe after many decades?) might be a re-establishment towards pre-dredging conditions. However, this recovery did not take place until now. A refill with muddy material at very low sedimentation rates seems to be the major problem, which was not predicted in the 1980s. However, maybe after many decades, the pits are flattened enough, that current velocity might be high enough to prevent an accumulation of muddy material. This might lead to a higher accumulation-rate of fine sand. However, we will provide the information that Pleistocene material cannot be re-stablished.

We will change the aims of this paper and the refill-processes will not be in the focus of this manuscript anymore (as also claimed by reviewer #1). We will try to set the objectives on the habitat change and the associated benthic communities.

Moreover, a re-framing of the introduction and the discussion (using more references, making more statistical analysis regarding a combination of the two datasets hydroacoustic and benthos analysis) is planned. This will be done by taking all your suggestion (e.g. PAH and the effect on foodwebs) into account.

**Specific comments in chronological order:**

1. **This paper discusses regeneration whereas it might be more applicable to use re-establishment.**

"Regeneration" will be replaced with "recovery" or "re-establishment".

2. **Line 29 – has reached a high level? Examples of the growth in extraction rates would be useful if it is considered that ongoing sand mining will pose a threat.**

Here are the numbers from recent ICES reports to reinforce this statement, which we will also add to the manuscript:

While between 1998 and 2002 approximately 53 million $m^3$ was extracted, a total of 73.2 million $m^3$ was extracted from the northern European Continental Shelf in 2018 (ICES, 2016, ICES, 2019).

**3. Line 36 – activities have led**

Ok, thank you. This will be revised.

**4. Line 54 - sonars allow the analysis of backscatter intensity – how? This needs more explanation.**

Sonar systems such as sidescan sonars allow to investigate the backscatter intensity by transmitting a hydroacoustic wave, which will be reflected by the seafloor and then received by the transceiver. The intensity of this reflection is used for seafloor classification by distinguishing between hard (strong backscatter response from the seafloor) and soft substrates (low backscatter response from the seafloor (Blondel and Murton, 1997; Blondel, 2003; Mielck et al., 2014, Mielck et al., 2015).

Further explanation will be added to the manuscript.

**5. Line 59 – poor English**

The sentence will be removed from the manuscript because it is a repetition.

**6. Line 64 introduces aim of paper – should be presented much earlier**

This can be done easily.

**7. Line 67 -will be used? Have been used.**

Yes, indeed. Can be revised.

**8. 53 grab samples for 55 x 5 km transects are not a lot of samples. A more detailed description of the sampling strategy should be presented to demonstrate the samples are representative of the different areas.**

The size of the study area is approx. 5 x 3 km. It would be possible to survey the whole study site using sidescan sonar with ~12 transects, since our sidescan sonar could measure with a swath range of ~250 m. The reason for the high amount of transects was the multibeam echosounder. It has only a small swath range in shallow water. In order to reach full-cover bathymetry, 55 transects were necessary.
Regarding the sampling strategy we already mentioned in the manuscript:
"The sampling positions generally followed a regular grid, but some positions were also selected on the basis of the bathymetric information in order to take samples both from the older dredging pits (older than 10 years) and the newer ones (see Fig. 4). "
The area and the occurring surface sediments are well known due to a prior study (Mielck et al. 2018). In the manuscript presented here, we aimed to describe the habitats including the benthic communities. We will add more details to the manuscript.

Mielck, F., Hass, H. C., Michaelis, R., Sander, L., Papenmeier, S., and Wiltshire, K. H.: Morphological changes due to marine aggregate extraction for beach nourishment in the German Bight (SE North Sea), Geo-Mar. Lett., 39(1), 47–58, doi:10.1007/s00367-018-0556-4, 2018.

**10. Line 99 - area was collected**

Revisable

**11. Some justification should be provided that all of the past mining pits are still visible. Were the locations compared to maps? How would you know if a pit was no longer visible?**

We did previous investigations, which focused on morphological changes due to marine aggregate extraction in this study area using bathymetric and seismic data between 1993 and 2017 (see above). Additionally, other investigations took place in this area (e.g. Zeiler et al. 2004). Moreover, the mining locations are well known by the coastal authorities (LKN-SH), so we can be sure that there are no complete refilled mining pits, which are not visible anymore.

We will clarify this in the revised introduction.

Zeiler M, Figge K, Griewatsch K, Diesing M, Schwarzer K (2004): Regenerierung von Materialentnahmestellen in Nord- und Ostsee. Die Küste 68:67–98.
https://doi.org/10.2314/GBV:599000627

**12. Benthos -line 159 – what is class 1 lower than? Class 0?**

Yes, it is lower than in class "0". We will rephrase the sentence like this:

After sand extraction, the macrozoobenthic abundance and species density was significantly lower in class "1" than in the unaffected area (class "0"), whilst after >10 years (class "2") only a minor increase became apparent when compared to class "1" (Fig. 5 and Table 1).

13. Line 169: Polychaete profited? Polychaete exploited suitable environment.

We will revise this sentence.

---

## Author Response (AR1)

**Review of manuscript bg-2020-17: "Persistent effects of sand extraction on habitats and associated benthic communities in the German Bight"**

**Reviewer #1:**

**General comments:**

This paper intends to investigate the effects of historic and recent intensive dredging on habitats and benthic fauna in the German Bight in a dredging area near Sylt. This is a follow-up paper of Mielck et al 2018 where the focus was on morphological changes due to sand extraction for beach nourishment. This definitely is of scientific value and interest in the field of effect studies and increasing demand of sand for both industrial purposes and coastal protection.

However, I had to find out myself that this was a 'follow-up' study and really needed to read the Mielck paper to get better insights in this study and understand the situation of the area. At least, this could have been better referred to. Furthermore, the way it is written and presented now, especially the discussion adds little value compared to the previous study. Although, in itself, it really is a different study and could add interesting new scientific insights.

But therefore, this manuscript has to be thoroughly reworked with focus on the new aspects i.e. defining the different habitats related to the dredging history of the sites and characterizing the benthic communities related to these habitats. The introduction should therefore at least make a clear referral to the previous study and the conclusions of that one.

 Moreover the manuscript should better introduce the available knowledge on the topic of impact of sand extraction on benthic habitats since too few references have been cited, while already quite some literature is available and this would situate the study in a broader perspective.

Objectives should also be better delineated to make clear what the main aim of this exact study was. This could also help maybe to explain the unconventional way of benthic sampling i.e. very small sample volume used for species identification compared to volume used for sediment analyses.

Results are too vague and too descriptive. Extra multivariate analyses should be done to characterize communities. Maybe acoustic data together with sediment data could be used in a PCA and these PCA results (=axis scores) can in their turn be used in the faunal analyses so that acoustic data are really used to determine benthic communities? This study would really benefit from a better combination of both datasets, since this is its strength. While in the current version of the manuscript, these two datasets are treated as separate entities.

Discussion is too superficial and adds very few new insights compared to the previous paper as well as said above. Plus it thus not really discuss the results of this study. I also do not agree with the conclusion made. The historic dredging actually caused a loss of habitat in my opinion. You even get a change in EUNIS habitat, so regeneration to the original habitat, without human intervention, will not be possible in that sense you cannot speak about regeneration/recovery. This could be discussed in the light of the MSFD, D6 seafloor integrity C1 habitat loss. See also specific comments for some extra input in the discussion that could lift it up to an interesting contribution for the scientific community. Also here, quite some literature is already available to put your results in a wider perspective but only very few references have been used. Looking into the existing body of literature

and putting your results in this wider perspective would give more body to the discussion. To conclude, the manuscript cannot be published in the current version, thorough revision is needed of all sections and some new analyses need to be done to make this a valuable contribution.

*Dear Reviewer #1,*

*Thank you for your revision and the numerous helpful comments and suggestions for improvement. Indeed, this manuscript is something like a follow-up to one of our previous studies (Mielck et al. 2018). The focus of that study was set on morphological changes due to marine aggregate extraction in the same study area using bathymetric data between 1993 and 2017. For the new study presented here, we collected new data and intended to focus on the impact of sand extraction on the habitats and the associated benthic communities.*

*You are right that the conclusion of the previous work needs to be better communicated in the introduction. We addressed this in line 105:*

*"This study is a follow-up to the previous study Mielck et al. 2018, which focused on morphological changes due to marine aggregate extraction in WDA using bathymetric data between 1993 and 2017."*

*Many thanks also for the provided literature and the hint to the ICES WGEXT reports, which were very helpful to improve our revised introduction with more facts and recent information about this topic, as well as for deeper discussion and a meaningful conclusion.*

*A better delineation of the aims of this study towards a combination of the used data sets (benthos analysis and hydroacoustic data) was a very good way to improve the whole study. Additional revisions and statistical analysis were done (see below). Table 2 was shifted to the supplementary material (as well as* SIMPER *and* ANOSIM*).*

*A recovery towards the original pre-dredging conditions is of course not possible, since the coarse Pleistocene sediment cannot be replaced without a new ice age. When sand mining started in 1984, the coastal authorities and also some scientist assumed the so-called "Wanderfeinsand" (migrating fine sand, Tabat, 1979; Köster 1979), which is ubiquitous in the German Bight, will refill the pits relatively quick. This has not happened until now due to weak sedimentation rates. However, a recovery to a fine-sand habitat might be possible (maybe in decades or centuries). When the pits are flattened enough, also current velocity (which decreased significantly in the pits) will increase again. This would prevent an accumulation of muddy material. We added these aspects to the manuscript (see below).*

**Specific comments in chronological order:**

Introduction

L30-31: 'current' with references from 2010 is somewhat outdated in my opinion. I would suggest to check ICES WGEXT reports where recent figures are yearly reported for NE Atlantic countries. Latest report has figures from 2018 even making a distinction between extraction for coastal protection and for industrial purposes, see https://www.ices.dk/community/groups/Pages/WGEXT.aspx

*The number has been corrected with data from the suggested ICES WGEXT report from 2019 and the references have been changed accordingly:*

*Line 35:*

*"For the northern European Continental Shelf, the extracted volume rose from some 53 million m$^3$ between 1998 and 2002 to a total of 73.2 million m$^3$ in 2018 (ICES, 2016, ICES, 2019)."*

L30-42: very few references while quite some papers have been published on these topics so would be good to support these lines with extra references. Just naming a few: Le Bot et al 2010, Foden et al 2009, Kubicki et al 2007, Van Lancker et al 2015, also in cooperative research report of ICES WG on extraction a lot of references are incorporated in chapter on ecological impact of sand extraction (http://ices.dk/sites/pub/Publication%20Reports/Cooperative%20Research%20Report%20(CRR)/CRR 330.pdf)

*The quoted references were expanded by the ones suggested above, as well as by others within the whole introduction.*

L64-66: refine/rephrase your objectives – maybe better in the form of hypotheses, research questions?

*This section has been thoroughly revised and is now presented earlier in the introduction.*

*Line 59:*

*"The aim of this study was to further determine the impacts of extensive marine aggregate extraction on the regional macrozoobenthic communities. The main objectives were to (i) gain a deeper understanding of the correlation between the prevailing habitats and the recovery state of the associated benthic assemblages, to (ii) evaluate temporal recovery patterns along with short- and long-term changes in the community structures and to (iii) investigate the potential of a re-establishment of pre-dredging conditions regarding fine sand domains (coarse Pleistocene material cannot be re-established because of weak current velocities). Therefore, dredging pits of different ages and, as a control, the sandy areas surrounding the extraction site were compared for sediment and benthic faunal composition. Using hydroacoustic gear and sediment grab samples, habitat maps were created combining sediment properties with information about abundance and diversity of the macrozoobenthos."*

L70 and further: this part should be moved to acknowledgements

*This part has been moved to acknowledgements.*

**Study area**

L74: make 'study area' a section under M&M

*We moved the section "study area" to "Material and Methods" (line 84).*

L75 - Fig1: Please include location of reference area(s) plus add information (best in a more detailed zoom) on e.g. geological layers, bathymetry and past and 'recent' dredged areas on the map, so it is more in line with the information provided in study area paragraph

*We revised Fig 1 and added bathymetric information, as well as the border of the Saalian moraine from which the sediment has been extracted. Additionally, we included the locations of differently aged extraction areas and the reference area. The zoom-factor was also increased.*

L78: water depths between 14 and 30 m, is this natural depth range or does this include extraction pits already? Confusing, I would suggest to report 'natural' depth

*The stated water depths include the extraction pits. The seafloor in the study area and also in the surroundings is very flat. The natural water depth ranges between 14 and 17.5 m in the research site.*

*We revised this in line 94:*

*"Natural water depths range between ~14 and ~17.5 m, while the pits left by sand extraction may reach down to 30 m water depth with diameters of approx. 1 km."*

L79: would be good to add cumulative amount of sand that has been extracted since 1984

*Since 1984, ~41 Mio m³ were extracted.*

*Revision in line 96:*

*"Since 1984, more than 40 million m³ sediment was extracted from this area using Trailing Suction Hopper Dredgers (LKN-SH, 2012, 2020)."*

L82: what type of dredging is done? Static or trailer dredging?

*Dredging was achieved using Trailing Suction Hopper Dredgers.*

*We added this information to the manuscript (see above).*

L83: typo add IS derived

*revised*

L84: 'prevails' strange wording, better 'takes place'?

*revised*

**Material and methods**

L94 'all-over' replace by 'over-all', 'high-resolute' replace by 'high resolution'

*revised*

L95-96: Please put transects and location of grab samples on a map.

*Transcts and positions of the grab samples are provided in the section 'results'. We added this information to the manuscript at line 112. (Captions of Fig. 2 and Fig. 4a were modified to highlight this information.):*

*Line 537:*

*"The exact positions of the taken grab samples lie in the middle of the pie charts;"*

*Line 529:*

*"Surveyed transects become visible as longish dark grey stripe proceeding in N-S direction."*

L95: These 55 transects were done for both multibeam and sidescan? Simultaneously or on different days? Please provide information on this in M&M section. Also not clear how long survey was, all in one week, several days throughout January? January can be quite heavy weather and shallow area so weather can have influence on measurements, certainly when spread over several days. Info needed.

*We added the missing information to the manuscript:*

*Line 109:*

*During the survey, which took place between January 25$^{th}$ and January 27$^{th}$ at calm weather conditions, multibeam echosounder and sidescan sonars were used simultaneously on all transects. Subsequently, 53 grab samples for grain-size and macrobenthic faunal analyses were collected on January 31$^{th}$.*

L111: please add what focus is of 330 kHZ and what of 1MHz sonar

*Information was added to the manuscript:*

*Line 130:*

*"Using different frequencies leads to more detailed information about the seafloor environment. Sidescan mosaics recorded with a low frequency generally yield information about large-scale objects on the seafloor (e.g. facies changes, sandwaves, megaripples), while a high frequency gives more information about small-scale structures such as ripple marks or stones (Mielck et al. 2015)."*

L126: very unconventional way of sampling benthos, very small samples for macrobenthos…I would expect the other way around big enough subsample for sediment and main sample for benthos? Why was this done this way? Clear justification is needed

*Initially, this study was planned as a follow-up to investigate the further sedimentation characteristics of the dredging holes. When deep sediment dredging was first applied, the deeper water layer and sediment inside the dredging holes temporarily depleted in oxygen and became close to azoic (Armonies & Buschbaum 2008, unpublished report for the national authorities). This was thought to be due to the diameter/depth aspect favouring stagnant conditions in relatively small but deep holes. As a consequence, the national authorities decided that further dredging should use the same holes, i.e. increase their diameter instead of creating many small holes. The present study was an opportunity to check whether or not the larger diameter of the dredging holes would facilitate water*

*circulation and thus enable permanent establishment of a macrozoobenthic community in the depths of the holes. Accordingly, the main questions to answer were:*

*(1) is there life macrozoobenthos in the dredging holes?*

*(2) if yes, typical sediment-specific (i.e. mud-dwelling) fauna or just temporary opportunists washed into the holes from ambient sediments?*

*(3) is the benthic biomass inside the holes already comparable to ambient sediments, i.e. is the function as a potential feeding ground for higher trophic levels already restored?*

*To answer these questions, small sediment cores taken along with the sediment ground-truthing samples were considered to be adequate. A full description of the benthic communities in- and outside the dredging holes was not intended in this study. Only after the present results, it is clear that the current state of macrozoobenthic development warrants further studies with focus on the benthic communities, and therefore, with a sampling design adequate to reveal far more details of benthic community composition.*

*Armonies, W. , Buschbaum, C. (2008): Fachgutachten Makrozoobenthos im Rahmen der UVS für das Sandentnahmegebiet "Westerland III" westlich von Sylt. Im Auftrag des Landesbetrieb für Küstenschutz, Nationalpark und Meeresschutz Schleswig-Holstein, pp. 1-94.*

L129-130: class 0 control, is this really control, undisturbed conditions?? What about indirect/secondary impacts? Can you be certain that these are not at all affected by the dredging?

*We know the seafloor west of Sylt very well. In Mielck et al. 2015, for example, we made a study in an area ca. 6 km northeast of the dredging area. Additionally, during the joint research project WIMO (2010-2015), we investigated several study areas more than 20 km south of it. The investigation shows, that the seafloor conditions are very similar to the conditions in class "0" (patterns of fine and coarse sand -> so called sorted bedforms). During the dredging process in our study area, only small portions of very fine material are released to the water. This fraction was not detected in the grain-size spectrum of our sediment samples in this area west of Sylt. It seems that the material is transported away by tidal currents into northern direction.*

*A direct impact due to dredging activities in class "0" can be excluded, since this is not permitted by the coastal authorities. In addition, the bathymetry does not show any signs of dredging.*

*Revision line 152:*

*"For statistical analysis, the sampling sites were classified according to their history of sand extraction: Class "0" with sites never dredged or indirectly impacted by sand extraction and thus serving as a control for undisturbed conditions; class "1" with the sites where sediment was extracted during the past 10 years;"*

Results

L140: replace 'excavation' by 'extraction' – try to be consistent throughout the manuscript

*Replaced*

L141: how do you know they are only partially refilled? What was depth after cessation of extraction? What is depth now? Please support your statements with numeric data.

*We showed the refill process in a previous paper about the dredging area (Mielck et al. 2018: Morphological changes due to marine aggregate extraction for beach nourishment in the German Bight (SE North Sea)). Here, we also used data from 1993, that showed water depth of ~26 m in the northern pit. At this point of time, dredging activities were still conducted in this part of the study area. In 2008, water depths were at ~22 m and 10 years later, no measureable differences in water depth occurred. Moreover, we made investigations with a seismic device that revealed old slope failures in the subbottom, which are an indicator for a refill process.*

*Additionally, data from Zeiler et al. (2004) reveal, that similar dredging depths (~ 33 m water depth) were also achieved in the year 1991. The dredging depth after cessation seemed to be generally at this level in order to limit the size of impact.*

*Zeiler M, Figge K, Griewatsch K, Diesing M, Schwarzer K (2004): Regenerierung von Materialentnahmestellen in Nord- und Ostsee. Die Küste 68:67–98. https://doi.org/10.2314/GBV:599000627*

*We added the reference (Zeiler et al., 2004) behind the phrase in line 169.*

L141-145: in text, you mention letters a, b, c , d but these are not indicated on figure2. Please make sure that your figures and text match.

*We modified the text and removed a,b,c,d.*

*Line 165:*

*"The hydroacoustic survey executed in January 2019 revealed that all of the past dredging pits are still visible by bathymetric lows down to 30 m water depth (Fig. 2 left, multibeam echosounder measurements) and the pits of the various periods are still distinguishable from each other. The pits in the middle part of the study area derived from dredging since 2017, the western ones from the 2009 to 2016 period, the southern ones from 1995 to 2008, and the northern ones from 1984 onwards (Fig. 2)."*

 – actually these are results from a previous study so delete here? Or clarify, since now it is confusing because you refer to/compare with earlier published study and description of these dredging pits is not the aim of this study."

*You are right, that the refill process is a result from the previous study. However, we think that it is important to show at least a bathymetric map of the study area and to give a hint to the refill process. Hence, we added a reference to Mielck et al. 2018.*

*Note: The bathymetry shown here was recorded in 2019 and was not published in Mielck et al. 2018.*

L146-147: please define what is high, intermediate and low backscatter

*The backscatter of the seafloor is illustrated in a grey-scale. When taking a look on Fig. 2 (right) you can see, that there are three different backscatter classes in the sonar-mosaic consisting of a range of grey values (high, intermediate, low backscatter). They represent three domains: coarse sand, fine sand, and mud. Since the backscatter in hydroacoustics is a complex process, it is not easy to define exact borders at the 8-bit color palette. There were many attempts in the past; however, the backscatter could be affected by very many parameters: e.g. the slope of the seafloor, the distance between the seafloor and the transducer, the slant range and angle of inclination, the gain-settings etc.*

*The post-processing software SonarWiz used in this study provides a function for automatic classification. However, there are always some artefacts in the resulting maps. Hence, we prefer to show the "raw" data in a sonar mosaic and do a supervised classification of the habitats with ARCGIS aided by our ground-truth data.*

*We changed this phrase in the manuscript, in order to clarify this aspect:*

*Line 171:*

*"The sidescan sonar measurements (Fig. 2, right) showed numerous features across the study area (Fig. 3). Ground truthing with grain-size analyses of the sediment (Fig. 4A and B) revealed that relatively high backscatter stands for coarse sand, intermediate for fine sand, and relatively low backscatter for muddy sediments. In addition, numerous stones were detected. Based on these sidescan sonar mosaics, the seafloor could be classified into four types (Fig. 3):"*

L156-157 – Fig4: would be good to have delineation of different dredging zones cf. old, new ones this would make interpretation of the maps more clear.

*These delineations are already included in Figure 2 (left). Fig. 4 already yields many features and we think that it is not a good idea to paint more lines in the figure. However, we put a hint in the caption regarding the ages:*

*Line 538:*

*"For age of the dredging zones cf. Fig. 2 (left)."*

L162: on which results this statement is based? Data in results are needed to support this? E.g. multivariate analyses or cluster analyses.

*We provided additional SIMPER and ANOSIM statistics/results. This information was added to the manuscript:*

*Line 159:*

*"Variations in community structure were analyzed by similarity percentage routine (SIMPER) and analyses of similarities (ANOSIM) procedures using the software package PRIMER 6 (PRIMER-E,*

*Ivybridge, U.K.). All benthos data and results of the statistical analyses are included in the supplementary materials."*

*Line 198:*

*"Paralleling the changes in sediment composition, the composition of the macrozoobenthic community strongly changed during the recovery phase (Supplementary material: Table 2). Compared to the ambient sediments, abundance of Magelona johnstoni, Pisione remota, Aonides paucibranchiata, Polygordius appendiculatus, and Goniadella bobretzkii, all sand dwelling species, sharply dropped in fresh (class 1) dredging holes (SIMPER, average dissimilarity 96.95%). Older (class 2) compared to fresh dredging holes show increases of abundance in mud dwellers such as Lagis koreni and Ophiura ophiura with its associate Kurtiella bidentata (SIMPER, average dissimilarity 96.38%; see Supplementary material Table 5). Both periods combined, faunal composition in older dredging holes is an assemblage of mud dwellers and strongly differs from the ambient assemblage of sand dwellers (average dissimilarity 90.94%). A community composition equivalent to ambient conditions was not reached in any of the extraction pits."*

L163-165: ? would think this is a result that should be under habitat mapping? This is ground truthing of your sidescan results

*Yes, indeed. We moved this part to 3.1 Habitat mapping.*

*Line 184:*

*"While undisturbed ambient sediments were mostly fine sands with intermingled patches of coarse sand, the bottom of the holes left by sand extraction was characterized by coarse sands that were rapidly covered by a layer of fine sand and later by muddy sediments. The increase in mud content in >10 year old pits was highly significant (Fig. 5A)."*

L166-170: unclear – is there a difference between undisturbed and control? First time mentioned in the paper. Please rephrase.

*No, there is no difference. We rephrased this throughout the whole manuscript.*

L174-177: idem as comment above, please use multivariate analyses to back up these statements with SIMPER to demonstrate which species are making the difference between the groups

We provided additional SIMPER and ANOSIM statistics to the result section (see above).

**Discussion**

L190: okay, low sedimentation rates but how does the mud comes in? Do these pits not act as traps for mud?

*Yes, this is correct. They act as sediment traps. The current velocity decreases inside the pits and muddy material, which is in suspension (most likely coming from rivers), accumulates in the pits. This is one result from the previous study (Mielck et al. 2018) and was also a finding from Zeiler et al.,*

*2004, who measured current velocities inside the dredging pits. We added this information to the manuscript.*

*Line 220:*

*"Finally, the strong decrease of current velocities inside the pits allow for sedimentation of suspended mud (Zeiler et al., 2004) turning the pits into mud areas. "*

L191-193: this is for the first time you mention the earlier study with which you compare the 2019 measurements – this causes the reader to be very confused all the way throughout the paper. Should be made clear from the beginning, even in introduction results of previous study should be situated.

*To set it in a better context, we added a hint to the earlier study to the section 2.1 Study area.*

*Line 105:*

*"This study is a follow-up to the previous study Mielck et al. 2018, which focused on morphological changes due to marine aggregate extraction in WDA using bathymetric data between 1993 and 2017."*

L212: is this a successional state? In my opinion, this is just an altered habitat which will never recover to the old state and reach a different equilibrium or has already reached it in the historic dredging pits. I would call this physical loss of benthic habitat (cf. MSFD descriptor 6 C1) due to dredging, even at the EUNIS level. For more background information see reports on this topic http://www.ices.dk/sites/pub/Publication%20Reports/Expert%20Group%20Report/Fisheries%20Resources%20Steering%20Group/2019/WKBEDPRES2/WKBEDPRES2_Report_2019.pdf + related info. Very weak discussion regarding the benthic results/benthic habitats. This is focus of the paper but it fails to add body to this topic while it actually should be the main part of the discussion. Combining the acoustic data with the biological data, is the interest of this paper.

*As already mentioned above: A recovery to a common fine sand habitat is possible, although the coarse sand is lost forever. When the pits are flattened enough, also current velocity will increase again. This would prevent an accumulation of muddy material.*

*However, your statements are very important and we hope that we added these aspects to our strongly revised discussion.*

*e.g. line: 249 and additional parts of the discussion:*

*"Currently, it is replaced by a transient habitat type with a shallow muddy surface layer on top of a sandy sub-surface layer. This allows some surface-dwelling mud-fauna to come in, but still excludes deep-dwelling mud-fauna such as Callianassa subterranea occurring in the muddy depression near the Island of Helgoland mentioned above. It may take some further decades of mud accumulation to reach habitat characteristics comparable to the Helgoland depression. The feeding-ground function for higher trophic levels may then be restored, but for a different set of users: predators limited to sandy sediments will be excluded while the larger water depth will limit profitability for others such as diving birds."*

*Further development into a typical mud community may take some more decades, until the mud layer has become thick enough for deep-dwelling species. This state may then remain for the next centuries, until the pits are largely backfilled and attain a surface sediment layer similar to original. But even then, living conditions may deviate from the surroundings, because the fine backfill sediments will persist deeper in the sediment. In addition, stones, gravel and coarse sand originally occurring at the sediment surface are unlikely to be replaced; without human interference (e.g. working with nature) their function as a habitat for epibenthic species is inevitably lost.*

L213-214: what are these habitat types? Where do you find which one? How related to dredging history? This is what should be discussed? Which habitat type, you found where and what are the indicator species for this type of habitat? As said above, this should be focus of discussion.

*Based on your suggestions, we revised the discussion. See above.*

L216: do you want restoration? Naturally, it will probably not be possible? So if you want restoration maybe mitigation through human intervention is needed? Or if not, leave it like it is, other suggestions?

See above (line 280).

What would be frequency needed for monitoring, yearly, every 5 years? Every 10

*During our investigations in the past years, we did a semiannual monitoring program in this area. The dredging season is generally between May and September. Before they started the extraction, we did measurements to evaluate the situation and what happened to the new dredging pits from the last season during winter storms. When the dredging season was finished in September, we did a second survey to detect the new dredging pits. Since the analysis of the benthic communities is a relatively great effort, we would do this only every two years in the future.*

*We added this Information to the manuscript:*

*Line 271:*

*"As a strategy to monitor the further development in the extraction sites, we suggest semiannual investigations of the occurring habitat types by hydroacoustic means combined with the analysis of the benthic communities every two years. "*

years? As you suggest, rate of infilling is very slow so why put money in a monitoring study where you already know the result? Wouldn't it be better to put the money in other research questions or mitigation measures or 'working with nature' designs? Or studies to prevent this happening again with the ongoing dredging? I am just putting forward some ideas, topics that could be included in the discussion and that would give it more scientific value. Now the discussion is too superficial, it could be lifted up by going more into depth in the main topics of your study.

*A regular monitoring is very meaningful, as the dredging activities will take place each year. We believe, hence, that these activities should be monitored by collecting hydroacoustic and benthic data on the same temporal scale. It will further allow us to control for other potential external influences,*

*such as storm events. However, we will come up with ideas, how the impact of dredging might be reduced.*

*When we started the monitoring program, it was the first time that sophisticated multibeam devices were used in this area. We already knew that the accumulation rate is very slow. However, we did not know that it is that slow. Additionally, we had no idea what happened to the benthic communities after more than 30 years after the impact.*

*The aspect "working with nature" was added to the manuscript.*

**Reviewer #2**

Review of manuscript bg-2020-17: "Persistent effects of sand extraction on habitats and associated benthic communities in the German Bight"

General comments:
This paper investigates the recovery of benthic invertebrate fauna following dredging for beach replenishment in the German Bight. The under lying premise of this work is that eventually the habitat condition will return to pre-dredging condition, and this will allow the re-establishment of predredging benthic fauna, which is found in nearby undisturbed areas. The author suggests that any benthic assemblage that differs from the unimpacted zones is a successional stage.

This premise has 2 flaws:
1. The author states that the sands being dredged are Pleistocene in age, and that the existing hydraulics of the system and sediment supply limits deposition to very fine sand and mud at very low rates of deposition. This information highlights that the hydraulics and sediment supply of the area has fundamentally changed since the sands were deposited in the Pleistocene, so it is difficult to understand how the author expects these conditions to be restored, even over many decades/centuries/millenia. In effect, Pleistocene marine conditions and sediment supply would need to be re-established for the pre-mining conditions to be re-established. What these changes do highlight is that the dredging is an unsustainable mining activity – e.g. the material being extracted will not be replenished, so the dredging is resulting in a permanent change in habitat conditions. The implications of this permanent change should be the focus of the paper.

*Dear Reviewer #2,*

*Thank you for you revision and your helpful comments.*

*You are right. Investigations show, that something like a new ice age with strong glacification and interglacials are necessary to reach pre-dredging conditions. However, this knowledge is relatively new regarding this study area. When sand mining started in 1984, it was assumed, that the so-called "Wanderfeinsand" (freely translated: moving fine sand, Tabat, 1979; Köster 1979) would refill the pits relatively quick. Of course, the Pleistocene material is gone forever, but a refill with fine sand would not indicate such a strong change in habitat characteristics when compared to muddy domains, since fine sand domains are relatively common west of Sylt. When you have a look at Figure 4 you can see, that the fine sand superimposes the Pleistocene material quite regularly. Hence, also a complete refill*

*with fine sand (maybe after many decades?) might be a re-establishment towards pre-dredging conditions. However, this recovery did not take place until now. A refill with muddy material at very low sedimentation rates seems to be the major problem, which was not predicted in the 1980s. However, maybe after many decades, the pits are adequately flattened, that current velocity might be high enough to prevent an accumulation of muddy material. This might lead to a higher accumulation-rate of fine sand. We provided the information in the strongly revised manuscript, that Pleistocene material cannot be re-stablished.*

*Line 242:*
*"Thus, these patches of hard substrata are inevitably lost for the benthic epifauna".*

2. The benthic assemblages present at the mined sites are not a successional stage that will ultimately lead back to the pre-mining assemblage. If the muds are not replaced by sand (which is highly unlikely given the quiescent hydraulics and limited sediment supply), then the mud loving assemblage will remain in perpetuity. The paper suggests that there is a successional order of benthic infauna, while at the same time saying the infauna reflects the sediment characteristics. Mud is not a successional stage to sand. The change in benthos due to the change in sediment, and why this matters, should be the focus of the discussion, not that it is an intermediate step leading back to the original assemblage.

This work should be re-framed to highlight the permanent changes that are occurring to the sediments, how the infauna has changed due to these impacts, and what are the implications of these changes. The author states that the sandy benthos is wide-spread and the mining is not a threat to the prevailing species – so the question is what are the implications, if any, of the conversion of sandy habitat to muddy habitat and the loss of sessile habitat? Discussing how these changes might affect other trophic levels or food webs, such as through the uptake of PAHs or other contaminants from the mud, would be more relevant than focussing on the (lack of) re-establishment of the original fauna.

It would also provide more context for the comment about monitoring PAH's which is otherwise unrelated to anything discussed in the paper.

*We changed the aims of this paper and the refill-processes is not in the focus of this manuscript anymore (as also claimed by reviewer #1). We set the objectives on the habitat change and the associated benthic communities. To emphasize this, we did further statistical analysis. Moreover, a re-framing of the introduction and the discussion (using more references, making more statistical analysis regarding a combination of the two datasets hydroacoustic and benthos analysis) was done. The impact on food webs was taken into account in the discussion: (e.g. regarding diving birds and predators).*

*Line 243:*
*The species composition of the macrozoobenthic infauna changed accordingly to the sediment composition in the dredging holes, once more demonstrating the well-known animal-sediment relationships deriving from the dynamic sedimentary and hydrodynamic environment (Snelgrove and Butman, 1994). Generally, recovery of the benthic fauna at disturbed sites depends on the recovery state of the sediment, and complete recovery is only possible, if the native sediment characteristics*

*are restored (Zeiler et al., 2004). Thus, complete recovery is only possible within the restrictions given above for the habitat characteristics. Until then, the original sandy habitat is lost for the benthic infauna, and thus as a feeding ground for higher trophic levels such as fish or diving birds. Currently, it is replaced by a transient habitat type with a shallow muddy surface layer on top of a sandy sub-surface layer. This allows some surface-dwelling mud-fauna to come in, but still excludes deep-dwelling mud-fauna such as Callianassa subterranea occurring in the muddy depression near the Island of Helgoland mentioned above. It may take some further decades of mud accumulation to reach habitat characteristics comparable to the Helgoland depression. The feeding-ground function for higher trophic levels may then be restored, but for a different set of users: predators limited to sandy sediments will be excluded while the larger water depth will limit profitability for others such as diving birds.*

*Revision regarding the impact of PAH:*

*Line 267:*
*"Besides the fauna, mud accretion in the dredging pits may also affect the chemical environment. Mud often shows enriched contents of polycyclic aromatic hydrocarbons (PAHs), chlorine hydrocarbons (Brockmeyer and Theobald, 2016) or heavy metals (Lakhan et al., 2003). In addition, hydrodynamic conditions allowing for mud accretion might also facilitate microplastic deposition. Whether or not the deep dredging pits may act as a sink for pollutants and whether or not the pollutants affect the benthic fauna remains to be studied."*

The introduction and presentation of the scientific question should be revised and strengthened, and the aim of the investigation should be clearly stated earlier in the paper. It would also be useful to provide more discussion about why deeper extraction pits might have a different recolonization trajectory as compared to shallower disturbances. Providing some hypothetical examples of how deeper disturbance could have different impacts as compared to shallow extraction would provide more context for the results. The paper would benefit from additional, and more recent references.

*We revised the whole introduction and specified the aims. Additional references were used for both the introduction and the discussion. Your suggestion regarding deep vs. extensive dredging operations was taken into account in our strongly revised discussion:*

*Line 256:*

*"Currently, sand mining is restricted to a relatively small part of the SE North Sea with vast surrounding areas with similar habitats and fauna. Because of deep dredging operations instead of extensive dredging, a vast habitat loss is therefore not expected and not a threat to all the sand-dwelling benthic species living in the area as it was the case in Italy for example (Varriale et al. 1985)."*

*Table 2 was removed to supplementary material.*

**Specific comments in chronological order:**

Introduction:

1. **This paper discusses regeneration whereas it might be more applicable to use re-establishment.**

*"Regeneration" has been substituted with "recovery" and "re-establishment".*

**2. Line 29 – has reached a high level? Examples of the growth in extraction rates would be useful if it is considered that ongoing sand mining will pose a threat.**

*Numbers from recent ICES reports were added to strenghten this statement.*

*Line 35:*

*"For the northern European Continental Shelf the extracted volume rose from some 53 million m$^3$ between 1998 and 2002 to a total of 73.2 million m$^3$ in 2018 (ICES, 2016, ICES, 2019)."*

**3. Line 36 – activities have led**

*The sentence has been corrected.*

**4. Line 54 - sonars allow the analysis of backscatter intensity – how? This needs more explanation.**

*Further explanation has been added.*

*Line 70:*

*"Sonar systems such as sidescan sonars allow investigating the backscatter intensity by transmitting an echo, which will be reflected by the seafloor and received by a transceiver. Backscatter allows to distinguish between hard/coarse (strong backscatter response from the seafloor) and soft/fine substrates (low backscatter response from the seafloor (Blondel and Murton, 1997; Blondel, 2003; Mielck et al., 2012, Mielck et al., 2015)) which is an additional parameter useful for seafloor classification."*

**Line 59 – poor English**

*The sentence has been removed because it was a repetition.*

**5. Line 64 introduces aim of paper – should be presented much earlier**

*This section was moved and revised:*

*Line 59:*

*"The aim of this study was to further determine the impacts of extensive marine aggregate extraction on the regional macrozoobenthic communities. The main objectives were to (i) gain a deeper understanding of the correlation between the prevailing habitats and the recovery state of the associated benthic assemblages, to (ii) evaluate temporal recovery patterns along with short- and long-term changes in the community structures and to (iii) investigate the potential of a re-establishment of pre-dredging conditions regarding fine sand domains (coarse Pleistocene material cannot be re-established because of weak current velocities). Therefore, dredging pits of different ages and, as a control, the sandy areas surrounding the extraction site were compared for sediment and benthic faunal composition. Using hydroacoustic gear and sediment grab samples, habitat maps*

*were created combining sediment properties with information about abundance and diversity of the macrozoobenthos."*

**6.   Line 67 -will be used? Have been used.**

*Revised.*

**7.  53 grab samples for 55 x 5 km transects are not a lot of samples. A more detailed description of the sampling strategy should be presented to demonstrate the samples are representative of the different areas.**

*The size of the study area is approx. 5 x 3 km. This is not very huge for hydroacoustics. It would be possible to survey the whole study site using sidescan sonar with ~12 transects, since our sidescan sonar could measure with a swath range of ~250 m. The reason for the high amount of transects was the multibeam echosounder. It has only a small swath range in shallow water. In order to reach full-cover bathymetry, 55 transects were necessary.*

*Regarding the sampling strategy we already mentioned in the manuscript:*

*Line 142:*
*"The sampling positions generally followed a regular grid but some positions were also selected on the basis of the bathymetric information in order to take samples both from the older dredging pits (older than 10 years) and the newer ones (see Fig. 1)."*

*The area and the occurring surface sediments are well known due to a prior study (Mielck et al. 2018). In the manuscript presented here, we aimed to describe the habitats including the benthic communities.*

*The following information was added to the manuscript:*

*Line 140:*
*"The surface sediments and morphology across WDA are already well-known from the prior study (Mielck et al. 2018) and were taken representative for all occurring seafloor environments."*

*Mielck, F., Hass, H. C., Michaelis, R., Sander, L., Papenmeier, S., and Wiltshire, K. H.: Morphological changes due to marine aggregate extraction for beach nourishment in the German Bight (SE North Sea), Geo-Mar. Lett., 39(1), 47–58, doi:10.1007/s00367-018-0556-4, 2018.*

**10. Line 99 - area was collected**

*revised*

**11. Some justification should be provided that all of the past mining pits are still visible. Were the locations compared to maps? How would you know if a pit was no longer visible?**

*We did a previous investigation, which focused on morphological changes due to marine aggregate extraction in this study area using bathymetric data between 1993 and 2017.*

*Additionally, other investigations took place in this area (e.g. Zeiler et al. 2004). Moreover, the mining locations are well known by the coastal authorities (LKN-SH), so we can be sure that there are no complete refilled mining pits, which are not visible anymore.*

*Zeiler M, Figge K, Griewatsch K, Diesing M, Schwarzer K (2004): Regenerierung von Materialentnahmestellen in Nord- und Ostsee. Die Küste 68:67–98. https://doi.org/10.2314/GBV:599000627*

*This information is now included in the manuscript:*

*Line 105:*
*This study is a follow-up to the previous study Mielck et al. 2018, which focused on morphological changes due to marine aggregate extraction in WDA using bathymetric data between 1993 and 2017.*

**12. Benthos -line 159 – what is class 1 lower than? Class 0?**

*Yes, it is lower than in class "0". We rephrased the whole section 3.2:*

*Line: 190:*

*"Sand extraction significantly changed macrozoobenthic abundance and species density, while there was no significant effect on biomass (ANOVA, Table 1). Scheffe post-hoc tests revealed that abundance was significantly lower in the dredged compared to the undisturbed sites (p<0.01 for the recently dredged sites and p<0.05 for the recovery sites), while there was no significant difference between recently dredged and recovery sites (p=0.53; Fig. 5A and B). After >10 years of recovery, the number of species returned to a level as high as for the control site (p=0.10), while the difference between undisturbed (control) and recently dredged sites was statistically significant (p<0.01). These changes macrozoobenthic species density and abundance were accompanied by significant changes in sediment composition, as exemplified by the percentage mud content which significantly differed between all combinations of disturbance classes (p<0.05; Fig. 5A)."*

*"Paralleling the changes in sediment composition, the composition of the macrozoobenthic community strongly changed during the recovery phase (Supplementary material: Table 2). Compared to the ambient sediments, abundance of Magelona johnstoni, Pisione remota, Aonides paucibranchiata, Polygordius appendiculatus, and Goniadella bobretzkii, all sand dwelling species, sharply dropped in fresh (class 1) dredging holes (SIMPER, average dissimilarity 96.95%). Older (class 2) compared to fresh dredging holes show increases of abundance in mud dwellers such as Lagis koreni and Ophiura ophiura with its associate Kurtiella bidentata (SIMPER, average dissimilarity 96.38%; see Supplementary material Table 5). Both periods combined, faunal composition in older dredging holes is an assemblage of mud dwellers and strongly differs from the ambient assemblage of sand dwellers (average dissimilarity 90.94%). A community composition equivalent to ambient conditions was not reached in any of the extraction pits."*

**13. Line 169: Polychaete profited? Polychaete exploited suitable environment.**

*Revised (see above).*

---

## Referee Report (RR1)

**Re-Review of manuscript bg-2020-17: "Persistent effects of sand extraction on habitats and associated benthic communities in the German Bight"**

The manuscript has improved a lot since the previous version. New analyses have been done and many parts have been rewritten or additions were made. I still believe this is a valuable addition to the literature on effects of sand extraction, certainly because it is such a valuable time series and a quite unusual way of dredging. However, I think there is still quite some room for improvement. First, the authors keep to their idea that return to pre-dredging habitat is possible and that what is present now is a transient habitat, which is not the case. Although they have nuanced this statement compared to the previous version, by saying it might take centuries, I believe that this is a local permanent change of the habitat from sand to mud/fine sediment (=loss of habitat). Although, it is just on a small surface area, this needs to be recognized throughout the whole manuscript and especially in abstract and discussion (see specific comments). What I am also still missing in discussion is the answer to the famous 'so what' question e.g. what are the implications for management of sand extraction in this area? What are implications of this changed habitat? What does it mean for future sand extraction? Also regarding monitoring strategy, discussion could be elaborated. The discussion needs this kind of paragraphs to get more body. Also the unconventional way of macrobenthos sampling should be recognized and discussed. Do you e.g. expect same results when conventional bigger samples would have been taken? This merits at least some discussion.

Thirdly, ANOSIM was said to be done, but I cannot find these results. Some kind of visualization in MDS plot or so would also help to show/read this results better. I also believe, the results are still lacking the combination of sediment and faunal analysis. This can be easily done in Primer and would strengthen the faunal results.

To allow publication, these different aspect should be tackled so reworking is needed. I leave it to the editor to decide whether this can be still be done in this journal.

**Some extra comments that I had whilst reading through the manuscript:**

Abstract:

L24-28: In my opinion, it should be recognized that there is habitat loss due to sand extraction. The habitat type has changed and will not recover without human intervention or a new ice age as matter of speaking. All evidence (morphological and faunal) is pointing in this direction. So, this should be stated this boldly in the abstract and not say that 'it will be a matter of centuries' and 'that it implies that coarse material is transported which actually will never be the case'… Just say it like it is without twists and turns. So rephrase this part of the abstract.

Introduction:

L36: is this the figure of coastal protection? Or total extracted? Since you are talking just about coastal protection best to mention this total figure…

L46-47:' even develop in unexpected directions' replace by 'or new habitats are developed'

L53: 'sensibility' should be 'sensitivity'

L64: aim 3 should be 'investigate the potential of a re-establishment TO pre-dredging conditions', all the other information is too much and contains already 'results'

Material & methods

L106: add 'for this study', hydroacoustic data and…

L112 and Fig 2: Position of VVgrabs should be put as dots on this map. Just refer to results is not sufficient. Putting them on this map helps to see the balance of your design.

L127: should be habitat character**istics**

L151: replace fresh by 'wet' weight, or even 'blotted wet' if you dipped the animals first

Results:

L187: was it not highly significant in the most recent pits? There are quite high mud concentration in the recent pits, it seems. L197: is this the same for the other sediment fraction as well? Or is it mainly mud that differs? Would be good to mention.

L200-205: no ANOSIM results are mentioned. It would be good to first tell where significant differences in species composition are situated. Eg between all classes or just between dredged and non-dredged? Plus actually, visualization of these results in an MDS plot would be good. Plus by combining sediment data with faunal data e.g. by vector overlay in your MDS plot and thus finding correlations between the axes and the sediment variables this would strengthen your results further.

Discussion

L233-234: you should not speak about restoration. I really believe this is not possible in this case. These pits might be filled naturally but will be a different habitat than the pre-dredging one. This is a permanent change.

L249: you cannot call this a transient habitat type, this is the current habitat type which has changed permanently, it is habitat loss due to dredging that occurred in this area. This is for the moment, still not enough recognized throughout the discussion.

L257: it is indeed not a vast habitat loss but it is a local habitat loss. The habitat has permanently changed. As said before this should be recognized.

L266-273: breaks flow of discussion why should this be mentioned? Delete? Better to focus on monitoring strategy instead, which comes suddenly out of the blue in the end. I also do not agree with semiannual surveys. This really is a waste of money for the old pits. Definitely since you have showed that backfill is so slowly. So there won't be big changes expected. In active dredging pits, it is another story, there it could be valuable to measure more often since there changes are ongoing. So important to make distinction between areas in monitoring strategy.

L282: situation will never be original.

L284-285: working with nature should be mentioned already in discussion or at least mention something about mitigation measures. E.g. stones/rocks could be placed in pits to restore these sessile epifauna when backfill has stopped?

Figures:

Fig2L528: add which are **'directly'** unaffected and replace 'dredged by 'dredging'

Fig 5: what do the letters a and b mean in the figure? Please add to the legend.

---

## Author Response (AR2)

**Reviewer #1**

**Re-Review of manuscript bg-2020-17: "Persistent effects of sand extraction on habitats and associated benthic communities in the German Bight"**

The manuscript has improved a lot since the previous version. New analyses have been done and many parts have been rewritten or additions were made. I still believe this is a valuable addition to the literature on effects of sand extraction, certainly because it is such a valuable time series and a quite unusual way of dredging. However, I think there is still quite some room for improvement.

First, the authors keep to their idea that return to pre-dredging habitat is possible and that what is present now is a transient habitat, which is not the case. Although they have nuanced this statement compared to the previous version, by saying it might take centuries, I believe that this is a local permanent change of the habitat from sand to mud/fine sediment (=loss of habitat). Although, it is just on a small surface area, this needs to be recognized throughout the whole manuscript and especially in abstract and discussion (see specific comments).

We have now taken this into account in the manuscript in a few places. Regarding the newly created mud habitats, we share the opinion of the reviewer that a permanent habitat loss has taken place. However, we (and also Reviewer#2) see slight signs of regeneration especially in the shallower pits in the northeast. Here, the sediment patterns that originally characterized the study area have already re-established themselves. Detailed information is provided below.

What I am also still missing in discussion is the answer to the famous 'so what' question e.g. what are the implications for management of sand extraction in this area? What are implications of this changed habitat? What does it mean for future sand extraction? Also regarding monitoring strategy, discussion could be elaborated. The discussion needs this kind of paragraphs to get more body.

Discussion was extended and revised in many places. For more details see below.

Also the unconventional way of macrobenthos sampling should be recognized and discussed. Do you e.g. expect same results when conventional bigger samples would have been taken? This merits at least some discussion.

The effect of the small sample size has been included in the discussion, see for line 159:

"For faunal analyses this sample volume may be unusually small but we judged it sufficient to find out whether dredging had a strong effect."

Thirdly, ANOSIM was said to be done, but I cannot find these results. Some kind of visualization in MDS plot or so would also help to show/read this results better. I also believe, the results are still lacking the combination of sediment and faunal analysis. This can be easily done in Primer and would strengthen the faunal results.

With respect to macrobenthos, both reviewers seem to expect far more than intended in this study: we regard this a pilot study just aiming to find out whether or not the re-filling of the pits with a sediment type finer than ambient has significant effects on macrofaunal species composition, abundance, or biomass. These effects indeed have been proven during this study by significant changes in total abundance, species richness, abundance changes of the dominant species, and hence a change in the dominance spectra. A more detailed analysis of the recovery process and all the factors included was not intended in this study, and our sampling design is not adapted to such advanced analyses. Thus, some questions raised by the reviewers are really interesting and should be

studied in more detail, but this requires extra sampling with a far more sophisticated design. An attempt to answer these questions from the current data base may succeed in some cases – as we tried with our ANOSIM analysis; however, a critical review of these results left the suspicion that the current data might be strongly over-interpreted by ANOSIM. Consequently, we dropped the ANOSIM and concentrated on changes in abundance of dominant species that can be proven without reasonable doubt.

To allow publication, these different aspect should be tackled so reworking is needed. I leave it to the editor to decide whether this can be still be done in this journal.

**Some extra comments that I had whilst reading through the manuscript:**

Abstract:
L24-28: In my opinion, it should be recognized that there is habitat loss due to sand extraction. The habitat type has changed and will not recover without human intervention or a new ice age as matter of speaking. All evidence (morphological and faunal) is pointing in this direction. So, this should be stated this boldly in the abstract and not say that 'it will be a matter of centuries' and 'that it implies that coarse material is transported which actually will never be the case'… Just say it like it is without twists and turns. So rephrase this part of the abstract.

We rephrased this part of the abstract while considering the arguments of the reviewer:

Line 23: "Since re-establishment of disturbed benthic communities depends on previous re-establishment of habitat characteristics, the low sedimentation rates indicate that a return to a pre-dredging habitat type with its former benthic community and habitat characteristics is likely to be impossible. Since coarse sand is virtually immobile in this area, a regeneration towards natural conditions is unlikely without human interference (e.g. mitigation measures like depositing coarse material on the seafloor to restore the sessile epifauna) or without a new ice age which once formed this area.

**Introduction:**

L36: is this the figure of coastal protection? Or total extracted? Since you are talking just about coastal protection best to mention this total figure…

This is the total amount of aggregates which were extracted. In the most cases, it was for coastal protection. We added the word "especially" to the sentence because there was also sand mining for other purposes:

Line 34: "marine aggregates needed especially for coastal protection"

L46-47:' even develop in unexpected directions' replace by 'or new habitats are developed'

Replaced and rewritten in Line: 46:

"Whether or not the benthic communities are able to recover to their pre-dredging state, remain disturbed, or new habitats with altered benthic communities are developed, is crucial information for a holistic assessment of the impact of such a coastal defense measure."

L53: 'sensibility' should be 'sensitivity'

corrected

L64: aim 3 should be 'investigate the potential of a re-establishment TO pre-dredging conditions', all the other information is too much and contains already 'results'

corrected

We rephrased the "aims" for more clarity:

Line 59: "The aim of this study was to further follow the re-filling process of the dredging pits and, as a new aspect, to find out whether and how extensive marine aggregate extraction affects regional macrozoobenthic communities. If local faunal composition was mainly ruled by larval supply, faunal composition inside the dredging pits may be similar to ambient sediments. Otherwise, if sediment composition was an important factor, faunal composition in the muddy sediments of dredging pits should considerably differ from composition in ambient sandy sediments."

Topic was moved to the discussion. The following sentence has been deleted.
"regarding fine sand domains (coarse Pleistocene material cannot be re-established because of weak current velocities)."

**Material & methods**

L106: add 'for this study', hydroacoustic data and…

Changed:
Line 111: "For the study presented here, hydroacoustic data and sediment samples were taken…"

L112 and Fig 2: Position of VVgrabs should be put as dots on this map. Just refer to results is not sufficient. Putting them on this map helps to see the balance of your design.

Yes, that makes sense. We put the positions of the grab sample stations on Fig. 2 (both bathymetry and sidescan mosaic). We deliberately chose not to label the stations because otherwise too much of the mapping would be covered.

L127: should be habitat character**istics**

Corrected in line 131

L151: replace fresh by 'wet' weight, or even 'blotted wet' if you dipped the animals first

Replaced by "wet" in line 157

**Results:**

L187: was it not highly significant in the most recent pits? There are quite high mud concentration in the recent pits, it seems. L197: is this the same for the other sediment fraction as well? Or is it mainly mud that differs? Would be good to mention.

We included a new version for Fig.5 showing more details of the sediment composition and added the new information (Tab. 2) to the manuscript:

Line 202:
"After >10 years of recovery, the number of species returned to a level as high as for the control site (p=0.10), while it was significantly lower in the recently dredged sites when compared to the undisturbed (control) site (p<0.01; Fig. 5). These changes in macrozoobenthic species density and abundance were accompanied by significant changes in sediment composition. Recently dredged sites had a low sand and a high mud content and were very heterogeneous sorted; older pit sediments were intermediate between fresh pits and ambient sediment, with intermediate sorting but with a mud content still far above ambient sediment level and with a median grain size only slightly higher than in fresh pits (Fig. 5). The percentage mud content differed significantly between all combinations of disturbance classes (p<0.05) and therefore, may be best suited as a proxy for changes in sediment composition.

Paralleling these changes in sediment composition, the composition of the macrozoobenthic community strongly changed during the recovery phase. Six of the ten most abundant species showed significant abundance variations between site classes (Table 2). Compared to the ambient sediments, abundance of the polychaetes *Magelona johnstoni, Pisione remota, Aonides paucibranchiata, Polygordius appendiculatus,* and *Goniadella bobretzkii*, all sand dwelling species, sharply dropped in fresh (class 1) dredging holes and did not return to ambient levels in older pits (Table 2). The polychaete Nephtys cirrosa temporarily vanished from fresh dredging holes while mud-dwelling Notomastus latericeus temporarily showed up in the fresh pits. Older (class 2) dredging holes showed increases of abundance above ambient level in brittle stars *Ophiura ophiura* and its associate bivalve *Kurtiella bidentata*. Finally, the trumpet worm *Lagis koreni* became the numerical dominant species in older pits (Table 2). Most of these changes are likely to be caused by the changes in sediment composition: species that correlate significantly with increasing median grain size decreased in the pits while the muddy-sand species *Lagis koreni* increased (Table 2). Thus, faunal composition in older dredging holes is an assemblage of muddy-sand dwellers and strongly differs from the ambient assemblage of sand dwellers. A community composition equivalent to ambient conditions was not reached in any of the extraction pits."

L200-205: no ANOSIM results are mentioned. It would be good to first tell where significant differences in species composition are situated. Eg between all classes or just between dredged and non-dredged? Plus actually, visualization of these results in an MDS plot would be good. Plus by combining sediment data with faunal data e.g. by vector overlay in your MDS plot and thus finding correlations between the axes and the sediment variables this would strengthen your results further.

Has been changed; see our comment on macrobenthos above.

**Discussion**

L233-234: you should not speak about restoration. I really believe this is not possible in this case. These pits might be filled naturally but will be a different habitat than the pre-dredging one. This is a permanent change.

We removed "restoration" and revised this part of the manuscript.

Line 245:
"Based on such low rates of sedimentation, a complete backfill of the pits is likely to take centuries (Mielck et al., 2018). After refill, the previous accumulations of muddy material in deeper layers of the sediment will persist, potentially affecting the living conditions for deeper-dwelling fauna.

This natural backfill cannot restore full pre-dredging conditions, for two other reasons: The first are differences in sediment composition. While coarse-to-medium sand was removed during dredging, the backfill material is fine sand with a high mud content. This is due to the relative immobility of coarse sand (Tabat, 1979; Werner, 2004; Mielck et al., 2015)."

L249: you cannot call this a transient habitat type, this is the current habitat type which has changed permanently, it is habitat loss due to dredging that occurred in this area. This is for the moment, still not enough recognized throughout the discussion.

We revised this part of the discussion:

Line 281:
"It is replaced by a new habitat type…"

L257: it is indeed not a vast habitat loss but it is a local habitat loss. The habitat has permanently changed. As said before this should be recognized.

At this point we have once again referred to the habitat lost
Line 287:
"Currently, sand mining accompanied by local habitat loss is restricted to a relatively small part of the SE North Sea…"

L266-273: breaks flow of discussion why should this be mentioned? Delete? Better to focus on monitoring strategy instead, which comes suddenly out of the blue in the end. I also do not agree with semiannual surveys. This really is a waste of money for the old pits. Definitely since you have showed that backfill is so slowly. So there won't be big changes expected. In active dredging pits, it is another story, there it could be valuable to measure more often since there changes are ongoing. So important to make distinction between areas in monitoring strategy.

We think, that the mentioning harmful chemicals which come along with the mud is an important aspect which should not be concealed (also in agreement with reviewer #2). We revised the discussion to avoid any "break flows". More detailed information is given below.
Regarding the monitoring strategy, we agree with the reviewer that it is not necessary to monitor the old pits twice a year because no strong changes are expectable.
We add these sentences to the manuscript:

Line 297:
"As a strategy to monitor the further development in the extraction sites, we suggest investigations of the occurring habitat types by hydroacoustic means combined with the analysis of the benthic communities. For younger pits with fast rates of change, this should be done twice a year for habitat types and every two years for benthic fauna; for older pits with a slow rate of change, a habitat survey every two years and a faunal analysis once per decade may be sufficient, to save money, time and resources."

L282: situation will never be original.

We rephrased the conclusion:

Line 314:
"This state may then remain for the next centuries, until the pits are largely backfilled and attain a surface sediment layer similar to original - at least regarding the morphology. But even then, living conditions may deviate from the former conditions, because the fine backfill sediments changed the habitat permanently."

However, in the Northeast, indications of regeneration become visible (in agreement with reviewer#2):

Line 318:
"However, at some positions, especially in the flat pits in the Northeast, slight regeneration towards pre-dredging conditions becomes visible. Here, patterns of sediment, which are coinciding with the seabed relief, recaptured the seafloor. This should be monitored in the future."

L284-285: working with nature should be mentioned already in discussion or at least mention something about mitigation measures. E.g. stones/rocks could be placed in pits to restore these sessile epifauna when backfill has stopped?

We mentioned these aspects in the discussion now. Additionally, discussion was extended.

Line 256:
"However, virtually no stones could be detected in the older dredging pits (Fig. 4 (c), (d)), as they were already buried by slope failures shortly after the dredging activity (Mielck et al. 2018). Thus, these patches of hard substrata and also the coarse sand areas are inevitably lost for the benthic epifauna. These habitats could only be restored by mitigation measures like depositing stones, gravel and coarse sand on the seafloor to allow for colonization of sessile epifauna."

Line 261:

"When planning the study, benthic fauna was included as an additional aspect in the pit recovery process because previous studies (Mielck et al. 2018) indicated that the sediments accumulating in the pits were finer than ambient sediments, and small subsamples for macrobenthos were deemed sufficient to check whether change in sediment composition had a strong effect on benthic macrofauna that would justify further studies. Significant differences of faunal composition between dredged sites and ambient sediments despite the small sample volumes indicates that changes are indeed strong, both on the community and on the species level. Larger samples might have proven this for even more species.

Species typical for coarse grained sand such as the polychaetes *Pisione remota*, and *Polygordius appendiculatus* could not re-establish in the fine sediments of the pits because they have an interstitial life-style equivalent to meiofaunal-sized organisms, i.e. they need a sediment type with pore sizes large enough for movement without displacing the sand grains. Based on the realized distribution across sediment types in the eastern North Sea, most benthic species seem to be restricted to a species-specific spectrum of sediment composition (Armonies, in prep.). However, since sediment composition correlates with many other factors such as hydrodynamic stress and sediment stability (Snelgrove and Butman, 1994), or oxygen supply and biogeochemistry (Giere 2008, Giere et al. 1988), the causes for these restrictions are not clear. Therefore, we can only state that the sediment types occupied in the dredging pits coincide with the sediment types occupied in the

surroundings (see supplementary material Table 2). In this sense, species composition of the benthic infauna changed according to sediment composition in the dredging holes."

Figures:

Fig2L528: add which are **'directly'** unaffected and replace 'dredged by 'dredging'

Added and changed

Fig 5: what do the letters a and b mean in the figure? Please add to the legend.

The figure has been extended and the legend revised.

**Reviewer #2**

The authors have gone a good length to improve the manuscript following constructive and detailed comments provided by reviewers 1 and 2. Improvements include a more elaborate description of the context of this study including previous work done in the area, a better definition of research questions for this study, better explanation of the methods, and more elaborate discussion on the longer term expectations regarding recovery to the original state at the sand extraction pits.

However, whilst both reviewers recommended to elaborate on the benthic communities established in relation to the different habitats formed as a result of dredging, which is probably the most interesting aspect of this study, the authors to my opinion could have done much better than the minimalistic 16 lines of text in results section 3.2 dealing with benthos, and statistical analysis merely addressing differences in benthos between more recent and past disturbance versus undisturbed state. In the discussion the authors state that the species composition of the macrozoobenthic infauna changed according to sediment composition in the dredging pits, but this is not substantiated by any direct analysis of fauna versus sediment characteristics.

Section 3.2 was strongly extended regarding this aspect. Supplementary material Table 5 was enlarged by more statistics (more details are provided below).

An underlying problem in the analysis of habitats is that habitat mapping based on hydroacoustics (illustrated in Fig. 2 and 4) seems to distinguish a much greater habitat heterogeneity in the dredging pits, comprising patches of coarse sand, fine sand and mud, than the uniform mud drape which is described in the text. To me, this gives the impression that the hydroacoustics often fail to distinguish the mud drape, but rather record the acoustically more reflective sand present below the mud. This would need to be properly acknowledged.

Sidescan sonars are normally not able to penetrate the sediment more than 2 cm. In the pits, the hydroacoustic backscatter is not easy to interpret because the soundings are reflected by the slopes resulting in high backscatter while the slope on the other side give no reflection because they lie in acoustic shadow. That's the reason why we marked the pits as "dredging marks" in the sediment distribution map (Fig. 4) and do not interpret the backscatter responses from the pits regarding sediment type. In order to analyze the sediment distribution in the pits, the results of the granulometric analyze were used. We mentioned this briefly in the caption of figure 3:

Line 568: "(D) cone shaped funnels representing the dredging marks on the seafloor. Since higher backscatter values occur on the slopes of the funnels that are directed towards the sonar source differences in sediment distribution are not distinguishable using hydroacoustic."

Regarding the hydroacoustically delineated habitats as shown in Fig. 4, no mention is made of seafloor relief in the study area except the dredging pits, even though a quick glance at the bathymetric and acoustic backscatter maps in Fig. 2 is sufficient to note a distinct coincidence of features of seabed relief and backscatter in the seabed areas surrounding the dredging pits. To me these features seem to represent low dunes composed of fine sand, lying on top of a coarse-grained Pleistocene sand which is outcropping in troughs in between the dunes. And the pattern of alternating dunes and troughs seems to extend also into the older dredging pits, especially in the northern part of the study area. This makes me wonder if the so-called "Wanderfeinsand" is actually moving across the shallower dredging pits, making habitats at these sites more dynamic than suggested by the authors.

We have now considered these aspects in the manuscript. Both the connection between bathymetry and sediment patterns and the slight signs of regeneration are now mentioned. More details below.

Regarding ground truthing of hydroacoustically delineated habitats, it does not help that the miniature pie charts presented in Fig. 4 are concealing the habitat present at the sampling location, and that some of the pies are overlapping.

We revised Figure 4. The pies are no more overlapping and the exact positions of the grabs are now not concealed by the pies as you suggest below.

In the attached document, I have included a large number of smaller mostly textual comments, and also some repeating of more general points of criticism raised above.

Line 25: replace "implies" by "requires"

Sentence was rewritten

Line 28: Working with nature implies making use of natural processes, but natural processes moving substantial amounts of sediment seem to be absent here.

"the term "Working with nature" was replaced by "mitigation measures":

Line 26:

"Since coarse sand is virtually immobile in this area, a regeneration towards natural conditions is unlikely without human interference (e.g. mitigation measures like depositing coarse material on the seafloor to restore the sessile epifauna) or without a new ice age which once formed this area."

Line 33: As a result, there is a world-wide high demand...

rewritten

Line 35: 2007 a or b?

It is "2007a". corrected (also in the references)

Line 35: Danovaro

Name corrected

Line 36: continental shelf

Corrected

Line 36: from 53 annually to 73 annually?

No, it was altogether 53 million m³ between 1998 and 2002. We added this to the manuscript:

Line 36:

"…the extracted volume rose from altogether 53 million m$^3$ between 1998 and 2002 to a total of 73.2 million m$^3$ in 2018."

Line 44: 2003

corrected

Line 46: 2007 a or b?

This is "b" – corrected (also in the references)

Line 58: 2005

corrected

Line 60: The statistical analysis addresses only relationship of fauna with disturbance class, not with different habitats or sediment characteristics determined in samples.

The chapter on animal-sediment relationships has been expanded in results, discussion, and supplementary material. See above (Reviewer #1) and below.

Line 65: hydroacoustics and sediment grab sampling

corrected

Line 68: Hydroacoustics have proven very effective for remote sensing seafloor classification

corrected

Line 72: In my understanding, the sonar transmits an acoustic pulse, the echo is the signal reflected from the seafloor.

That is true. We have corrected this in the manuscript:

Line 72:

"Sonar systems such as sidescan sonars allow to investigate the backscatter intensity by transmitting an acoustic pulse, which will be reflected by the seafloor and received by a transceiver."

Line 76: 2017

corrected

Line 76: Also sediment properties? What other purpose does groundtruthing as mentioned in previous sentence have?

Yes, indeed. We have deleted the superfluous sentence.

Line 78: Two things that require more thorough analysis: 1) relationship benthos with sediment type, 2) relationship sediment type with acoustic backscatter

1) has been changed, see above

2) Determining the relationship between sediment type and acoustic backscatter is not any easy task. Because of alternating gain-settings, slopes on the seafloor, range from the seafloor to the transducer etc. the backscatter intensities can strongly shift. In our case especially regarding the deep pits, this would statistically not be possible. For interpretation of sidescan sonar data, it is usual to do it in a "supervised way". Unsupervised methods often lead to a wrong classification of the seafloor (e.g. BSH, 2016: Anleitung zur Kartierung des Meeresbodens mittels hochauflösender Sonare in den deutschen Meeresge-bieten. BSH Nr. 7201).

We added this sentence to the manuscript:

Line141: The results of the grain-size analysis were used to relate the backscatter intensities to the prevailing sediment distribution.

Line 89: outcropping in the WDA

We added this to the manuscript in Line 90.

Line 90: No outcrops of boulder clay here? Or are glacial deposits actually pushed up fluvial sands and gravel originating from NW European rivers?

WDA is crossed by a Saalian moraine called PISA. The coarse sand and the stones are relicts of these moraine which was – of course – strongly eroded during sea level rise in the Holocene. Fine sand was transported through rivers and partially accumulated in this area. Boulder clay was not found here.

Line 91: From Fig. 1 it seems the moraine core strikes WNW. Or do the grey lines merely indicate the boundaries of where Pleistocene sediment is outcropping from below Holocene sand?

No, the grey lines show the position of the moraine core (after Köster 1979). Vast parts of this core are now covered with fine sand as you can see in Fig. 4. However, you are right. In our study area, the moraine strikes more in WNW direction. Further to the west, the direction changes to NNW. We changed this in the manuscript.

Line 92: "At the study area, this moraine core strikes in west-north-west direction…"

Line 92: Also fine grained sand! Please give more detail on dimension and orientation of these bedforms. Fig 2 seems to show low sand dunes with ~100 m wavelength and E-W strike, perpendicular to coastline of Sylt, which alternate in regular pattern with shallow troughs in which coarse sediment prevails. Fig. 3 shows very regular linear megaripples with ~1 m wavelength and N-S strike, parallel to coastline. Both types of bedforms would suggest active bedload sediment transport.

The origin and development of the sorted bedforms was subject of the investigation by Mielck et al. 2015. They can also be found further to the north were the study took place:

Mielck, F., Holler, P., Bürk, D., and Hass, H. C.: Interannual variability of sorted bedforms in the coastal German Bight (SE North Sea), Cont. Shelf Res., 111, 31–41, 2015.

We added some further information regarding size, wavelength and setup to the manuscript:

Line 93: "The surface of the seafloor in WDA is characterized by bands of coarse-grained rippled sand, so called sorted bedforms which are very common west off Sylt (Diesing et al., 2006; Mielck et al., 2018). Most of these bands have a wavelength of ~100 m and strike in east-west direction. The ripples within the coarse-grained areas do not strike in the same direction as they were most likely

formed during storm events (alignment perpendicular to storm direction). The sorted bedforms are often overlaid by a layer of migrating fine sand that leads to the consequence that their shape is frequently altered (Mielck et al. 2015)."

Line 92: Bedforms is not a term specifically reserved for bands of coarse-grained rippled sands, but for any sedimentary structure resulting from sediment transport.

That's true, however "sorted bedforms" is a fixed term in the coastal geology. Sometimes these structures are also called "rippled scour depressions", however, sorted bedforms is more common.

Line 96: No capitals

corrected

Line 98: This would be the best place to specify the dates of the recent and past dredging zones as given in Fig. 2.

Information was added to this place:
Line 103:
"The study area includes recent dredging zones (younger than 10 years), sand deposits exploited already more than 10 years ago, and…"

Line 99: The older pits are shallower than the more recent ones. Does this reflect a change in dredging practice, of were the older pits originally deeper than they are now? If so, how much deeper and what does this tell about backfill?

All these questions were already answered in Mielck et al. 2018. The dredging practice never changed. Originally, the old pits were as deep as the new ones. However, after a short time (~1 year), the pits flattened because of slope failures at the steep slopes. Later mud accumulated in the pits.

Line 100: I would expect that in the steep sides of the pits the same coarse Pleistocene sand is present as was extracted from the pits? Or do you mean fine sand that entered over the rim of the pit and temporarily accumulated on the sloping sides of the pit before moving further down?

We have expressed this somewhat misleadingly. Of course, there is coarse sand at the pits. However, during the flattening process, this sand slides to the bottom of the pits and fine sand, coming from the immediate vicinity (seafloor surface) covers the slopes and also the bottom of the pits.
For more clarity, we changed this part in the manuscript:

Line 105: "The Pleistocene coarse sands in the pits exposed during sand extraction were rapidly covered by a layer of fine sand due to slides at the steep slopes. The fine sand originates from the immediate seafloor surface around the pits."

Are Pleistocene sands not naturally outcropping off Sylt? Only after removal of Holocene top layer?

We meant, of course, the Pleistocene sand within the pits. Outcropping of Pleistocene material can often be found off Sylt (e.g. sorted bedforms). See above.

Line 100: were

corrected

Line 101: accumulated

corrected

Line 105: by

corrected

Line 106: collected

corrected

Line 106: Meaning new hydroacoustic data, in addition to data collected earlier between 1993 and 2017?

Yes. We changed this sentence for more clarity:

Line: 109: "For the study presented here, hydroacoustic data and sediment samples were taken using the research vessel *Alkor* in January 2019."

Line 136: Please explain if delineation of habitats as shown in Fig. 4 was exclusively done on the basis of hydroacoustic data, after which ground truthing was used to link the different backscatter signatures to sediment type. Or if ground truthing data was also taken into account in delineation of the habitats.

While delineation of the habitats using ArcGIS, also the ground truthing data (i.e. mean grain sizes after Folk and Ward calculated with GRADISTAT) were used. We added these sentences to the manuscript:

Line 140: "The size of the stones was determined by measuring slant angle and length of the acoustic shadow using the software EdgeTech Discover. The results of the grain-size analysis were used to relate the backscatter intensities to the prevailing sediment distribution."

Line 141: as

corrected

Line 141: and for macrobenthos analysis

Added to the manuscript
Line 146:
"For ground truthing of hydroacoustic data and for macrobenthos analysis, a total of 53 grab samples were taken…"

Line 149: Reviewer 1 asked for clarification of the small size of the benthos subsample, in comparison to available total sample. This is not really answered here.

Small sample size is explained in the methods section:
Line 155: "For faunal analyses this sample volume may be unusually small but we judged it sufficient to find out whether dredging had a strong effect."

Line 151: wet

revised

Line 152: In view of the heterogeneity of the sediment in the study area, the relationship between fauna and sediment type seems also very relevant to investigate!

This part has been considerably expanded (see above and below).

Line 153: for

corrected

Line 154: for

corrected

Line 154: for

corrected

Line 160: analysis

corrected

Line 167: produced by

corrected

Line 169: Do you have data on the original depth of the older pits, and how much sediment has filled in since cessation of dredging?

This question was also already answered in Mielck et al. 2018. Originally, the old pits had a depth of ~ 30 m (below sea surface). Now, they have a water depth off approx. 25 m. However, in the beginning, the accumulation rates were much higher because of slope failures. Today, we assume sedimentation rates (with mud) between 2 and 18 mm. We added this information shortly to the manuscript:

Line 173: "Thus, even the oldest depressions have only partially refilled with sediment after 35 years (quick backfill of about 5 m and very low sedimentation rates after the first year; c.f. Zeiler et al. 2004; Mielck et al. 2018)."

Line 171: There is much overlap in content with lines 174-181. However, no mention whatsoever of seabed topographis features that are very distinctly coinciding with backscatter patterns: low dunes composed of fine sand extending in E-W direction, alternating with shallow troughs in which coarser or more  acoustically reflective sediment is outcropping. These E-W oriented features are most clearly developed on the undisturbed seabed surrounding the dredging pits, but seem to extend also into the older and shallower pits, especially in the north of the study area. This would suggest more mobility of the Wanderfeinsand than assumed in this manuscript.

We have included the connection between topography and backscatter in the manuscript and also mentioned the mobility of the fine sand dunes:

Line 177: "Ground truthing with grain-size analyses of the sediment (Fig. 4A and B) revealed that high backscatter domains represent rippled coarse sand zones (sorted bedforms). Intermediate backscatter stands for fine sand. Coarse and fine sand zones were often demarcated by sharp borders (Fig. 3A). These backscatter patterns distinctly coinciding with the topography (compare Fig. 2 left and right), while low dunes composed of relatively mobile fine sand (extending in east-west direction) alternating with shallow troughs where relative immobile coarser sediment is exposed. These seafloor features are most pronounced on the undisturbed seafloor around the dredging pits, but also appear to extend into the older and shallower pits especially in the North of the study area."

Line 175: Outside the dredging pits, it seems that high-backscatter and intermediate backscatter domains alternate in a regular pattern, distinctly coinciding with alternating E-W oriented shallow troughs and low crests. This coincidence between backscatter and bathymetry should be mentioned. With the additional information that low-lying high-backscatter domains are characterised by coarse sand with stones, and the higher ground intermediate backscatter domains by fine sand, suggests the presence of low dunes or sheets of fine sand that migrate in along-shore direction over a surface covered with lag deposit of coarse sand with stones. Has this been discussed in the Mielck et al 2018 paper? This pattern of high-backscatter troughs and intermediate backscatter crests seems to extend also into the older dredging pits, in particular the northernmost one, which would suggest that large-scale bedforms may be migrating across these pits.

The topography was extensively discussed in Mielck et al 2018 and also the sediment distribution (18 grab samples were taken), however not the backscatter intensities from the different domains since there were no sidescan sonar mosaics available in this study. We have included the connection between topography and backscatter in this manuscript (see above line 177).

Line 175: high backscatter

changed

Line 175: Is high backscatter related to the coarse nature of the sediment, or also or even mostly to the fact that the sloping sides of the ripples are oriented towards the sonar source?

High backscatter is a combination of both coarse-grained sediment and the slopes. When the ripples are orientated perpendicular to the movement direction of the sonar source, they might not be visible in the sonograms, however, the backscatter will still be high. But this is only very rarely the case, as the ripples almost never run dead straight.

Line 177: The meaning of light/dark shades should better be explained in the figure caption.

The caption of figure 3 was revised:

Line 565: (A) rippled coarse sand (dark) and smooth fine sand (bright) demarcated by a sharp border. (B) cobbles and boulders. In the direction of the sonar source, these objects initially exhibit high backscatter followed by a bright acoustic shadow from which no backscatter can occur. (C) very smooth mud area with low backscatter surrounded by a domain of fine sand with intermediate backscatter. (D) cone shaped funnels representing the dredging marks on the seafloor. Higher backscatter values occur on the slopes of the funnels that are directed towards the sonar source.

Line 180: This is also particularly true for the conspicuous feature in the north of the study area near 54 deg 56 min N, which again coincides with a distinct step in bathymetry. Is this a man-made feature?

No, this is not a man-made feature. This is part of the moraine (see Fig. 1). At this position, coarse Pleistocene sediment is outcropping and not covered by the "Wanderfeinsand".

Line 181: , in the sonograms represented as areas with uniform low backscatter,

The sentence was supplemented
Line 188:
"Extended areas of mud, in the sonograms represented as areas with uniform low backscatter, could only be…"

Line 182: funnels? cavities? depressions?

We changed it to „depressions".

Line 184: I would rather call it a sediment distribution map and a bathymetric map.

changed

Line 185: This is not a very adequate description of what seems a distinct pattern.

We modified this sentence:
Line 193: "While undisturbed ambient sediments were mostly fine sands interspersed with strips of coarse sand."

Line 186: In this phrasing, nothing reflects the patchy and to some extent patterned distribution of coarse and fine sands that is also seen in the pits, and very limited occurrence of mud. Are the hydroacoustics unfit to detect relatively thin drape of mud covering the bottom of the pits, and merely recording the underlying acoustically reflective sand units? If this is the case, this should be properly acknowledged!

Sidescan sonars are not able to penetrate the sediment more than 2 cm.

In the pits, the hydroacoustic backscatter is not easy to interpret because the soundings are reflected by the slopes resulting in high backscatter while the slope on the other side give no reflection because they lie in acoustic shadow. That's the reason why we marked the pits as "dredging marks" in the sediment distribution map (Fig. 4) and do not interpret the backscatter responses from the pits regarding sediment type. In order to analyze the sediment distribution in the pits, the results of the granulometric analyze were used.

Line 186: If anything is clear from Fig. 5A, it is that mud content is significantly higher in the younger and deeper pits than in the older and shallower pits, which is opposed to the notion that the pits are gradually filling in with mud as time passes.

Right, we clarified that "pure" mud accumulation was temporary in young pits and turned to 'muddy fine sand' accumulations in older pits. Figure 5 was revised and Table 2 was added.

Line 189: Only 16 lines describing relationships between benthos and habitat/sediment type along with development of the dredging pits through time is a bit disappointing. Most of this merely addresses relationship between benthos and disturbance class, not habitat.

Section 3.2 was strongly extended regarding this aspect.

Supplementary material Table 5 was enlarged by statistics on the realized niches with respect to sediment composition of numerically dominant species in ambient sediments of the study area. These data show that the species identified as sand dwellers all attain highest abundances in sand and had rarely or never been recorded in mud. Conversely, *Lagis koreni* was scarcely found in sandy sediments but attained maximum abundances in mud; hence it was classified a mud-dweller.

Line 190: And what about sediment type?

has been rephrased (see section 3.2).

Line 215: coinciding with seabed relief features

Revised:
Line 231: "Before the dredging activity started in 1984, the study site was characterized by patterns of fine and coarse sand (sorted bedforms, coinciding with seabed relief features), which are very common in this area…"

Line 216: And seem to extend into the older and shallower dredging pits.

We also believe that slight tendencies towards regeneration can be seen in some places. For Reviewer#1 this is denied for the mud habitats. Here also no signs of regeneration are to be recognized and one can therefore speak of a habitat loss.

Revision in line 233:
"These pre-dredging conditions are still present between the dredged areas and east of them (Fig. 2, 3A, 4) and seem to extend into the older and shallower dredging pits especially in the Northeast."

Line 218: While dredging was ongoing the bottom of the pits was littered with stones that were too large to be sucked in by the dredger.

Revision:
Line 235: "The dredging pits, in contrast, have different surface layers. Directly after dredging, the surface is composed of coarse sand and stones, that were too large to be sucked in by the dredger."

Line 219: From backscatter maps it seems that the fine sand cover is patchy in the pits, revealing also the underlying coarser sand.

This process takes longer so that at the beginning stones and coarse sand can be observed on the seabed. After a few months, however, this is no longer the case, as the taken sediment samples also show.

We modified the sentence to be more precise:
Line 236: "Soon afterwards, this layer got more and more covered by fine sand probably deriving from the (formerly steep) rims of the pits."

Line 219: Or entering across the rims?

Maybe, when the slopes are flattened enough. However, this is not easy to prove with our data.

Line 220: allowed

Corrected

Line 221: But according to what is shown in Fig. 2 and 4, mud cover is far from uniform over the bottom of the pits, but restricted to a few small patches.

Since the sidescan sonar recordings in the fresh pits are too patchy for classification (see above) we marked these areas as "dredging marks" in Fig. 4. When you take a look on the sediment samples coming from this area (e.g. samples 28, 29, 31 and 33), you can see the high mud content within these pits. Maybe after a couple of years when the pits have flattened, the mud will also become visible in the sidescan sonar.

Line 239:
We added "after a couple of months" to this sentence in order to show that this process needs some time.

Line 224: This only shows fill up rate at later stage. Any data on initial fill up rate?

In the beginning, the fill-rates are relatively high due to slope failures. We mentioned this in line 173 and line 236. However, the backfill should not be in the focus of the manuscript because it was already discussed in Mielck et al. 2018.

Line 231: According to lines 89-90 the extracted sand is not just sand but coarse sand.

That's true. It is coarse-to-medium sand. We changed this in line 250.
"While coarse-to-medium sand was removed during dredging…"

Line 233: If the coarse sand is a Pleistocene relict, and backfill is by fine sands which are mobile, then I don't see how coarse sands could become mixed in the backfill.

At request of reviewer #1, this part of the manuscript has already been revised.

Line 244:
"Based on such low rates of sedimentation, a complete backfill of the pits is likely to take centuries (Mielck et al., 2018). After refill, the previous accumulations of muddy material in deeper layers of the sediment will persist, potentially affecting the living conditions for deeper-dwelling fauna. This natural backfill cannot restore full pre-dredging conditions, for two other reasons: The first are differences in sediment composition. While coarse-to-medium sand was removed during dredging, the backfill material is fine sand with a high mud content. This is due to the relative immobility of coarse sand (Tabat, 1979; Werner, 2004; Mielck et al., 2015)."

Line 234: What previous accumulations of fine material?

See above

Line 240: year?

2019, revised

Line 243: This is not demonstrated by any direct statistical analysis, only inferred from differences in fauna from different disturbance classes.

Additional data are given in the supplementary material substantiating the classification of dominant species with sediment types. The discussion section was strongly extended.

Line 248: The replacement of sandy by muddy seabed does not necessarily mean loss of feeding ground. According to Fig. 5D, biomass hasn't changed significantly. Actually, the trapping of organic detritus in the pits could be expected to lead to higher benthic production. The increased water depth in the pits seems a more likely barrier to predation of benthos by diving birds.

The potential loss of feeding ground has been refined: it's a potential loss of feeding ground for depth-limited predators and/or for those feeding on sand ground only.
As regards the relation of organic matter supply and benthic production, we fear the relation is not as simple as expected by the reviewer; potential oxygen depletion with increasing organic matter content has to be taken into account, in particular in a low-circulation environment as are the pits.

Line 262: Not demonstrated in this study.

The correlation of sediment composition and faunal composition has been refined in both the results section and the discussion, and with references in supplementary materials.

Line 267: Indeed, by favouring deposition of organic detritus, boosting benthic productivity.

See above comment to line 248: right, organic matter deposition may boost benthic productivity – but only if organic matter was in short supply before AND if increased organic matter content does not result in oxygen deficiency.

Line 272: Even if Wanderfeinsand is supposedly practically immobile, and deposition rate of mud in the pits is far below detection of hydroacoustics?

Yes, that is right. We modified this part of the manuscript. The monitoring should be limited to the fresh dredging pits and the ongoing mining.

Line 297: "As a strategy to monitor the further development in the extraction sites, we suggest investigations of the occurring habitat types by hydroacoustic means combined with the analysis of the benthic communities. For younger pits with fast rates of change, this should be done twice a year for habitat types and every two years for benthic fauna; for older pits with a slow rate of change, a habitat survey every two years and a faunal analysis once per decade may be sufficient, to save money, time and resources."

Line 373: Not referred to in text.

deleted

Fig. 3: Refer to Fig. 2 for explanation of ship tracks and different shades of grey. And some more explanatory text would be welcome.

We added more explanatory text to the caption of Fig. 3. Line 565:

"Figure 3: Seafloor features detected within the two sidescan sonar mosaics: (A) rippled coarse sand (dark) and smooth fine sand (bright) demarcated by a sharp border. (B) cobbles and boulders. In the direction of the sonar source, these objects initially exhibit high backscatter followed by a bright acoustic shadow from which no backscatter can occur. (C) very smooth mud area with low

backscatter surrounded by a domain of fine sand with intermediate backscatter. (D) cone shaped funnels representing the dredging marks on the seafloor. Higher backscatter values occur on the slopes of the funnels that are directed towards the sonar source."

Figure 4: Unfortunately, the pie conceal the nature of the sediment patch from which the sample was taken: fine or coarse sediment or mud. Better shift the pie's a bit sideways with tieline connected to exact sampling point. Also the overlapping of the pie's for samples 49-54 is not really convenient.

We revised Figure 4 as you suggested. The pies are no more overlapping and the exact positions of the grabs are now not concealed by the pies. Tielines were also added.

Figure 5: Please explain meaning of a and b

Fig. 5 was replaced by a new version and the legend revised

Why on a log-10 scale? The range of values can be easily displayed on a non-log scale.

We revised Figure 5 and removed the log-10 scale (though this didn't change anything).

---

## Author Response (AR3)

Dear authors,

I have read through the reviewers' assessments and your answers to their questions. I believe your manuscript is now fit for publication in BG. Going closely through your revised manuscript, I have a number of suggestions to improve your manuscript.

Sincerely,

Lennart de Nooijer

Dear Mr. de Nooijer,

thank you for your quick response and the suggestions for improvement. Please find our revision below.

Best regards

Finn Mielck

Abstract
line 25: '… characteristics is unlikely.'

Changed

line 26: please change 'natural' into 'original' or 'pre-dredging' or something.

We used the term "pre-dredging" and added an "also" to the sentence, because the word "unlikely" already occurs in the previous sentence.

Line: 27/28: I suggest to remove the idea that a new ice age may restore the pre-dredging situation. It is not wrong, but irrelevant for this paper. I know that reviewer #1 hinted on this, but upon reading the abstract, I found it a out of place (at least here).

I agree with you. We removed the ice age from the manuscript.

Introduction
Line 63: '…differ considerably from those…'

changed

Material and Methods
Figure 1: I think the scalebar in the inset can be removed.

We removed the scalebar in the inset.

Line 147: 'Van Veen grab'
corrected

Results
Caption Figure 3, last line: 'hydroacoustics'

corrected

Line 201: 'the dredged area compared...' (I assume)

Indeed. We added "area" to the sentence.

Line 215: N. cirrosa in italics

Changed to italics:

*"Nephtys cirrosa"* and also *"Notomastus latericeus"* in the next line.

Discussion
Line 236: replace 'more and more' by 'increasingly'

replaced